# communications
# engineering

# Cognitive gripping with flexible graphene printed multi-sensor array

Tania Mukherjee[1] & Dipti Gupta [1]✉

Robotics for task simplification of domestic, household, workplace and other assistive activities require efficient robots with decision-making capabilities. Here we report a fully printed graphene-based capacitive multi-sensor array (CAPSENSAR) employed in a cognitive robotic gripper (COGBOT) for decision-making operations. The CAPSENSAR created a contactless capacitive impression of the gripped object surface to determine the optimum gripping pressure. The controlling unit of the COGBOT was associated with an algorithm to address potential breakage. If slippage was detected via the array, the grip pressure was revised to reduce the possibility for damage. This facilitated slippage-free and damage-resistant gripping of the target objects without user interference. Array fabrication was straightforward using a customizable electrode design with cost-effective and biocompatible materials.

[1] Plastic Electronics and Energy Laboratory (PEEL Lab), Metallurgical Engineering and Material Science, Indian Institute of Technology Bombay, Mumbai 400076, India. ✉email: diptig@iitb.ac.in

As modern robots continue to evolve through advanced engineering and smart technology, their scopes of utilization are no more confined to execute specific user defined task in heavy industries[1–3] like automobile, shipping, etc., packaging sector[4,5], pharmaceutical industry[6], medical assistance and healthcare services[7–12], and other sophisticated and dedicated fields[13–16] but has shown their usefulness to execute varied task in domestic, office, and household chores. Researchers and engineers have introduced robots for simple and light-weight tasks[16–19] such as cleaning, cooking, targeted object delivery, and transfer of objects over short distances for domestic and household purpose although they are still expensive and large, which limits their commercial success[20]. Such robots may play an important role in the todays fast lifestyle by enhancing the quality and wellness of life and also serve as an assistive help for people with restricted or no mobility as in paralytic patients, amputees and in old or aged people.

To diversify robotic applications in the household, at workplace and as medical assist, the robots must be reliable in use, portable, cost-effective, low powered and user friendly. Researchers developed a robotic compliant jaw gripper with three fingers that executed pickup and drop operations, manipulation of objects at the fingertips and conformally grasping large and irregular objects between the three fingers[21]. Each finger consisted of plurality of interconnected phalanges configured to grasp an object. Due to the multiple end-arm tool and numerous phalanges at each finger required for conformal grasping, this gripper possessed complex mechanical design and included multiple sensors integrated at the jaws, which made the gripper expensive, non-user friendly, and unsuitable for domestic and light-weight economic operations. Soft grippers were increasingly investigated for the design of lighter, simpler, and more universal grippers. Engineers developed soft two fingered pneumatic grippers with integrated stretchable electro-adhesive actuators, which could pick-and-place flat and flexible materials and also delicate objects such as a light bulb[8]. Despite being an intelligent and shape-adaptive handling system, the gripper design involved complexities due to mechanical architecture and integrated electronic circuitry. Moreover, the gripper failed to estimate the shape and size of the target object and thus needed continuous human cognizance for successful operation. To enhance the grasp on the target object, researchers designed gripper jaws based on a set of trapezoidal jaw modules that maximized the contact between the jaws and the object at its desired final orientation as it grasped and avoided jamming[22]. These jaws were constrained to grasp and rotate the object to its desired orientation and achieved a conformal grasp on the object. However, such jaws may easily damage delicate objects in the process as the optimum gripping force, which was dependent on the contact locations, were not continuously measured. To ensure safe gripping of delicate objects and to promote user friendliness in light-weight operations, automatized robots were developed to assists humans in various tasks. Such robots utilize human involvement by remotely controlling pickup and drop operations and object manipulations. However, these automated robots are often at a risk of malfunctioning especially when the robots are not programmed to deal to small variations in the target operation, which may only seem to be a trivial task for humans. Designing and programming a robot with such human cognitive adaptability can be expensive, complex, and involve complexities in mechanical design and electronic circuitry[4,21,23]. Collaborative robots termed as 'cobots' operated alongside human involvement and used human cognizance to complete a given task effectively[13,24]. However, such robot was not suitable for various domestic, household and office task as it required continuous and appropriate involvement of human cognizance and failed in the purpose for automated

pickup and delivery tasks, which in turn limited its acceptability. Since robotic applications involved simplification of task for improvement in human lives, highly efficient robots with cognitive decision-making capabilities, will transform and diversify their utilization in, domestic, household, workplaces and other assistive tasks. Thus, to provide a user-cognizance-free automated approach for smart robots we propose the development of ink-jet-printed graphene-based capacitive multi-sensor array (CAPSENSAR) and utilized for cognitive decision-making tasks in cognitive robotic gripper (COGBOT), ensuring slippage-free and damage-resistant gripping of the target object.

**Key features of CAPSENSAR and COGBOT.** The proposed COGBOT utilized capacitive sensor array (CAPSENSAR) comprising of proximity and pressure sensor arrays integrated on the pair of end-arm tool of the gripper and a programmable controlling unit to achieve cognitive gripping of target objects. The CAPSENSAR was fabricated with non-overlapping duplex graphene-printed electrodes array on either side of bendable polyimide dielectric/substrate of thickness 30 μm, offering highly sensitive capacitive proximity sensing[25] in the range 3 cm and offered 3 times higher dynamic range than the previously reported capacitive proximity sensors[26–29]. The advantage of the printed electrode based capacitive proximity sensor as compared to the optical[30], inductive[31], and ultrasonic[32] type proximity sensors lies in its flexible architecture, due to which it can be conformed on the end-arm tool of the robotic gripper for face landscape estimation[33]. The graphene-based-printed electrodes are most suitable for bendable devices as the graphene micro-flakes in the electrode shears over each other under local strain due to bending and restores back on withdrawal. Moreover, the graphene lattice can withstand a localized strain upto 20% without suffering damage and hence provides excellent bendability to the graphene-based-printed electrodes. The capacitive pressure sensor array in the CAPSENSAR constituted a duplex graphene electrode architecture where the Ecoflex rear cladding served as the elastomeric dielectric layer and facilitated measurement of pressure ($P$-) contour of the gripped region of the target object and hence determined the optimum pressure $P_{grip}$ for secure gripping. This would help to prevent slippage and deformation of a given object[34–37] during gripping. The use of CAPSENSAR on the end-arm tool of the gripper facilitated the detection of position, determination of the face landscape of the object and cognitive determination of its pair of suitable flattest faces for gripping followed by measurement of the optimum gripping force for gentle and firm gripping and monitoring of slippage and damage pattern in the object in the post gripping stage. These features in the COGBOT allow the reduction of mechanical complexities through the design of simple end-arm tool in the gripper to perform difficult tasks. The electrical complexities in the COGBOT were reduced through the design of CAPSENSAR where the capacitive proximity and pressure sensor units communicated with the controlling unit using the same feedback circuitry. The array-based design of the proximity sensor facilitated the generation of three-dimensional landscape of the target object's face, which helped in the estimation of position and alignment of the opposite faces of the object between the pair of end-arm tool of the gripper. In the gripping stage of operation, the same array operating as the pressure sensor generated a three-dimensional pressure landscape of the gripping area on the object's face and therefore detected any slippage or damage of the object due to gripping. With the help of a pair of CAPSENSAR integrated on the end-arm tool of the gripper, it was capable of detecting the opposite pair of flattest face of an irregular target object for gripping. Using this cognitive data, the COGBOT was

able to grip the pair of flattest face of the target object effectively, thereby eliminating the need for the design of complex end-arm tool for executing precise tasks. The controlling unit of the COGBOT was associated with programmable anti-slippage and anti-damage algorithm that prevented the system from undesirable mishandling and deformation of the object, respectively. This slippage-free and damage free execution was achieved through a sequence of steps involving the (a) measurement of the optimum gripping force for a given object and (b) continuous monitoring of variations in pressure landscape of the object under gripped state. The determination of gripping force prior to gripping and thereafter gripping of the object with the measured optimum force provided a cognitive automatic approach that led to a promising technology that operated without human cognizance. The continuous monitoring of the pressure landscape through iterative cycles provided a self-regenerative process of force optimization to prevent slippage and damage. Such cognitive based automated approach for effective gripping eliminated human error, actions or reflexes during operation. The COGBOT offered a low power consumption of 3 W, as it used a low power driver motor for the robotic arm and low power CAPSENSAR for executing cognitive tasks. Since the CAPSENSAR calibration curve for proximity sensor is dependent on the material of the object, the COGBOT needs to be recalibrated for a specific material for shape and size estimation[38,39].

**Architecture of CAPSENSAR and the design of COGBOT.** The CAPSENSAR consists of duplex architecture of top (TE) and bottom (BE) graphene electrode array on either side of thin polyimide (PI) sheet of thickness $d_{PI} = 30\,\mu m$, together with rear Ecoflex cladding of thickness $\delta = 1\,mm$ covering the BE and 300 nm thick polyvinyl alcohol (PVA) as the encapsulation layer on the TE as shown in Fig. 1a. The graphene-based TE and the BE arrays are ink-jet printed on either surface of PI layer to form a bendable capacitive architecture with PI layer as the dielectric. The TE consisted of an $(i \times 1)$ array of parallel electrodes. The BE consisted of an $(1 \times j)$ array of such electrodes and were aligned mutually perpendicular to that of the TE. The TE and BE on top

and rear sides of the PI sheet form an $(i \times j)$ array of proximity and pressure sensor units where $i = 5$, $j = 4$. Each electrode in turn consists of linear arrangement of square shaped sub-electrodes (sE) of area $3 \times 3\,mm^2$, which are connected to the neighboring sE through interconnects of length 3 mm and width 1 mm. The sE of TE and BE are aligned mutually perpendicularly on either side of the PI dielectric layer such that they constitute non-overlapping electrodes of a capacitive architecture forming $(5 \times 4)$ array of fringing field capacitive proximity sensor (Supplementary Fig. 1). The working principle of the CAPSENSAR is described using mathematical modeling (Supplementary Discussion 1 and supplementary Fig. 2) and supported by COMSOL simulation (Supplementary Discussion 2). Theoretical results showed that the distortion in fringing electric field $E_{fr}^{obj}$ in presence of a target object obeys an arctan relation with proximal distance $z$ and was confirmed using simulation results in supplementary Fig. 3a. This was illustrated in supplementary Fig. 3b, c, which show the change in $E_{fr}^{obj}$ in the vicinity of TE and BE. The $E_{fr}^{obj}$ between the TE and BE changes as the object gradually approaches the plane of the CAPSENSAR (Supplementary Discussion 2.1, supplementary Fig. 4), thereby changing the measurable output capacitance $C_{out}^{i,j}]^{pre}$ of the $(i, j)$ sensor unit. All $(i,j)$ sensor units of the CAPSENSAR cumulatively generate capacitive impression of the exposed face of the object in the form of $(i \times j)$ capacitive matrix of that object face and can be transformed into $(i \times j)$ proximity $(z)$-matrix (Supplementary Discussion 2.2) or the $(i \times j)$ pressure $(P)$-matrix (Supplementary Discussion 2.3) using suitable calibration equations based on the regime of CAPSENSAR operation. This capacitive impression cumulatively obtained from $(i, j)$ proximity sensor units generated a capacitive landscape of the exposed face, which was used for shape detection of a given object, as illustrated using simulation studies of fringing electric field $(E_{fr})$-array impression in Supplementary Fig. 5a, b, and c for geometric objects sphere, cone and disc (Supplementary Table 1), respectively. This face detection feature of CAPSENSAR was illustrated using COMSOL

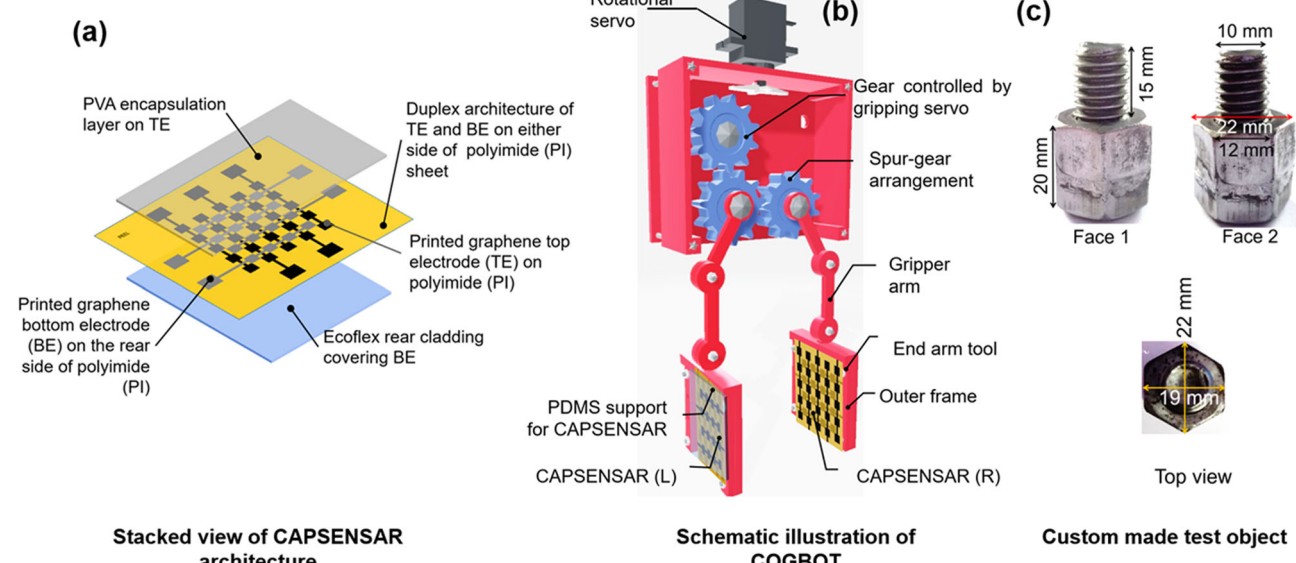

**Fig. 1 Components of COGBOT and different exposure faces of custom-made test object. a** Architecture of the CAPSENSAR showing the different layers of PVA, PI bearing the graphene TE and BE and the rear Ecoflex layer. **b** Front view of COGBOT with left and right CAPSENSAR integrated on the gripper palms (end-arm tools). **c** The Hex nut and bolt arrangement as the custom object used in the demonstration of cognitive gripping, showing the orientation and dimensions of exposure Face 1, Face 2, and top view.

simulation investigation for different exposed faces of custom-made object (supplementary Fig. 5d) and showed distinct dissimilarities in $E_{fr}$ array impression for different exposed faces (supplementary Fig. 5e, f) of the object. The proximity sensor array of the CAPSENSAR offers a high $z$-axis resolution of 0.091 mm for $z < 30$ mm and $x$–$y$ flatness resolution of 3 mm. On the other hand, the PI sheet bearing the TE and BE array and the elastomeric dielectric Ecoflex rear cladding constitutes $(5 \times 4)$ array of pressure sensors. The variation in output capacitance $C_{out}^{i,j}]^{pre}$ due to change in Ecoflex thickness $\delta$ under applied pressure was investigated using COMSOL simulation (Supplementary Fig. 6) and the results were utilized to measure the gripping force to be applied on the target object. The CAPSENSAR is spanned over an active sensing area of $37 \times 31$ mm with an overall thickness of 1.03 mm. The fabrication method of the CAPSENSAR is described in supplementary Discussion 3 where the fabrication process flow is schematically illustrated in supplementary Fig. 7. A robotic gripper was 3D printed using Acrylonitrile Butadiene Styrene (ABS) where two identical as-fabricated CAPSENSARs were integrated onto the left and right end-arm tools of the gripper to construct the COGBOT (Supplementary Discussion 4). The schematic diagram of the COGBOT is shown in Fig. 1b. The detailed schematic illustration of different components of the COGBOT is provided in Supplementary Fig. 8.

## Results and discussion

**Shape and size estimation of objects.** For efficient performance of the COGBOT, the performance of the CAPSENSAR as proximity and pressure sensing array needs to be investigated. A separate experimental set-up (as described in Supplementary discussion 5.1 and illustrated in Supplementary Fig. 9a) was designed to characterize the $(i, j)$ elementary sensor unit of the CAPSENSAR. The electrical circuitry for the characterization of CAPSENSAR was described in Supplementary discussion 5.2. The experimental results obtained from electrical measurements of proximity and pressure sensor units are described in Supplementary discussion 5.3. The proximity sensor unit operated in the dynamic range of $z = 0$–120 mm and showed enhanced sensitivity of 0.012 mm$^{-1}$, low response time of 0.3 s and high stability with error 3.5% within $z = 0$–30 mm as depicted in supplementary Fig. 10a, b. On the other hand, the pressure sensor unit operated in the dynamic range $P = 0.5$–5 kPa, with a dead band region below $P = 0.5$ kPa. It offered a response time of 0.4 s and stability with error 5.3%, and provided a sensitivity of 0.006 kPa$^{-1}$ as shown in Supplementary Fig. 10c, d, respectively. The proximity sensor array was utilized for the estimation of three-dimensional face landscape of the target object's face exposed to the device as described in Supplementary discussion 5.4 while the pressure sensor array determined the area of contact and hence the estimation of $P_{grip}$ on the object during gripping. The capacitive outcomes from all the $(i, j)$ sensor units in the CAPSENSAR generated capacitive impressions in the form of $(i \times j)$ matrix. These capacitive impressions were used for the estimation of three-dimensional (3D) face landscape of the exposed face of different objects as illustrated in supplementary. Supp. Fig. 11a, b, and c for geometric test objects such as sphere, cone and disc, respectively. The CAPSENSAR holds promise for contactless shape estimation of an unknown object when this capacitive matrix was transformed into $z$-matrices and respective gridded $z$-contour representations as illustrated for the same set of objects as shown in Supp. Fig. 11d, e, and f. The size estimation was performed using the two-dimensional profile of the exposed faces of the same objects in the $x$–$y$ plane as shown in Supp. Fig. 11(g, h), (i, j), and (k, l). Similarly, the capacitive matrices were converted to $P$-matrices and $P$-contour plots when the CAPSENSAR was operated in pressure sensing mode.

**Cognitive operation of COGBOT.** In the previous section, CAPSENSAR was established as capacitive device capable of non-contact face landscape detection and pressure contour generation when in contact with the object. These features were utilized in the design and development of the robotic gripper capable of cognitive operations. The operation of COGBOT was demonstrated with custom-made nut and bolt arrangement of mass $(m) = 65$ g with dimensions as shown in Fig. 1c. This object has six rotational and six reflection symmetries with two distinct exposure faces denoted as Face 1 and Face 2. The computational steps in cognitive gripping by COGBOT are illustrated through an algorithm as shown in Fig. 2. The COGBOT was constructed by integrating a pair of CAPSENSAR on both the end-arm tools (palm) of the gripper. The pair of CAPSENSAR was connected to the Arduino Mega microcontroller (MC) through separate 8:1 multiplexer (MUX). The circuit diagram of the COGBOT is shown in Supplementary Fig. 9b. The rotation and the gripping servos communicate with the MC through Signal 1 (SIG1) and Signal 2 (SIG2), respectively. The Arduino Mega board is powered by the computer. The operations of the COGBOT was divided into four major steps as: (1) Initialization for cognitive operations, (2) Determination of fittest pair of opposite flattest faces for gripping, (3) Pressure sensitive intelligent gripping with optimum pressure $P_{grip}$ and (4) Slippage resisting gripping, as described below.

The pair of gripper palms with integrated CAPSENSAR were placed at an initial proximal distance $z = 10$ mm from the edges of opposite faces of the object such that the pair of CAPSENSAR and the object was pre-aligned along the central axis of rotation before the commencement of operation. Initially, the pair of gripper palms was aligned along the direction of largest dimension of the object to ensure unobstructed capacitive scanning of different faces of the object and this angular position was taken as the initial alignment angle $\theta = 0°$ for the pair of gripper palm arrangement of COGBOT (Supplementary Fig. 12a). Once the pair of opposite flattest faces of the object was ascertained using separate $z$-contour plots obtained from CAPSENSAR (R) and CAPSENSAR (L) attached to gripper palm 1 and 2, respectively, the COGBOT reorients the pair of gripping palms along the direction normal to the plane of flattest faces for convenient and secure gripping of the object (Supplementary Fig. 12b). The procedure for the determination of the opposite pair of fittest gripping faces of the object is described below.

The pair of fittest gripping faces of the object was obtained by analyzing the respective three-dimensional (3D) $z$-contour plots for different faces of the object. To obtain the 3D $z$-contour plots for different faces of the target object, the MC instructed the rotation servo to rotate the arrangement of the pair of robotic palms with the CAPSENSARs about the central axis of rotation by incremental angles of 30° executing a total angle of 150° in six subsequent steps as shown in Fig. 3a. Thus, the parallelly aligned arrangement of $(i \times j)$ proximity sensor arrays of the two CAPSENSAR (L, R), recorded capacitive impressions of the pair of mutually opposite faces of the object at alignment angles $\theta = 0°$, 30°, 60°, 90°, 120° and 150°, where the corresponding positions were represented by Pos. A-a, Pos. B-b, Pos. C-c, Pos. D-d, Pos. E-e, Pos. F-f, respectively, as shown in Fig. 3a, b. For a fixed alignment angle $\theta$, the CAPSENSAR (L) and CAPSENSAR (R), recorded capacitance impression of corresponding faces of the target object and were expressed in terms of their respective $z$-matrices in MC using the calibrating expression of supplementary Eq. 15 for respective $(i, j)$ sensor units.

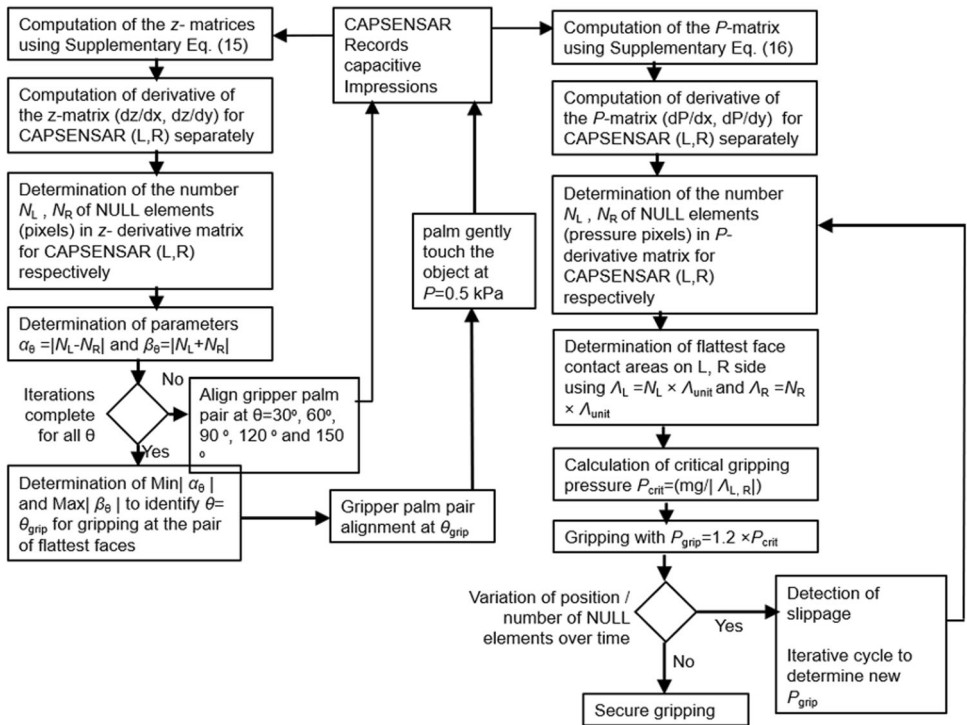

**Fig. 2 Algorithm for the cognitive gripping by COGBOT.** Illustrative representation of the algorithm used for identifying the convenient pair of object faces for gripping with optimum pressure $P_{grip}$.

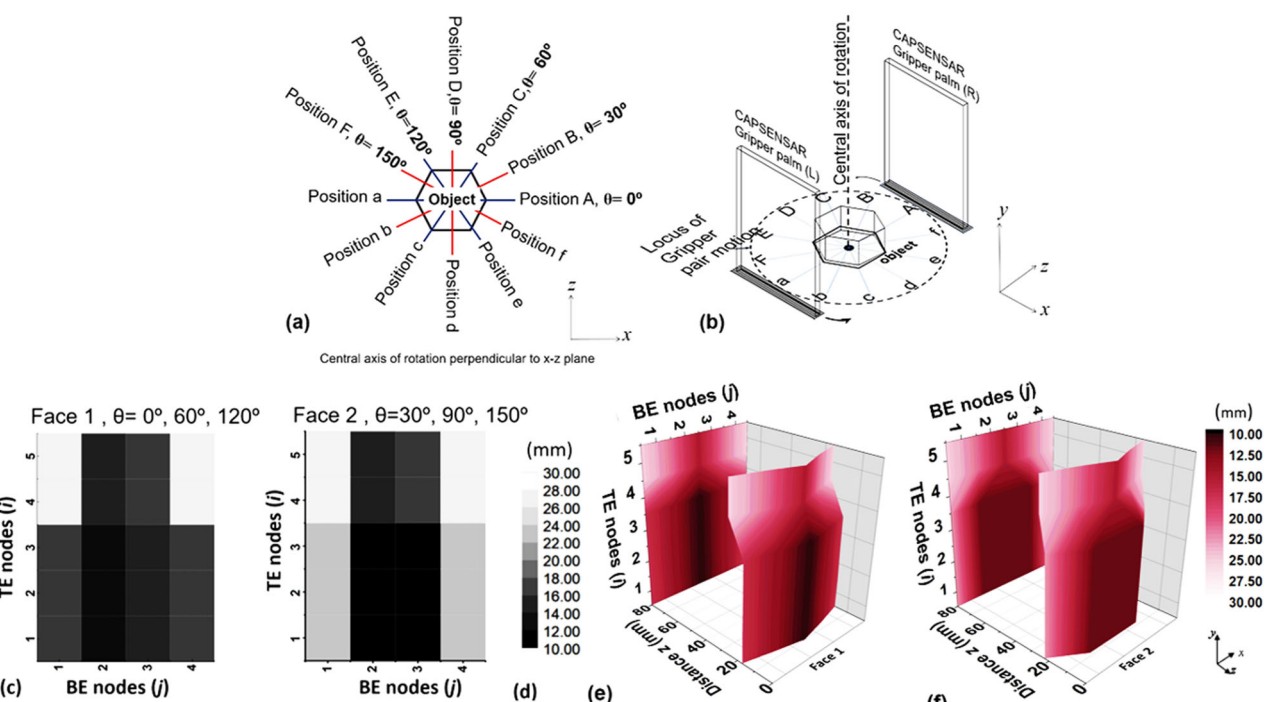

**Fig. 3 Detection of the pair of opposite flattest face of the object.** Schematic representation depicting **a** six different scanning orientations at Positions. A-a, B-b, C-c, D-d, E-e, F-f for alignment angles $\theta = 0°$, 30°, 60°, 90°, 120° and 150°, respectively, and **b** the locus of the movement of pair of parallel palm arrangement about the central axis of rotation around the object, Black and white representation of $z$-matrices obtained from ($i \times j$) sensor units for exposed faces. **c** Face 1 and **d** Face 2 of the object at different $\theta$. Gridded 3D $z$-contour plots for **e** Face 1 and **f** Face 2 of the custom object acquired by the pair of CAPSENSARs at different $\theta$. Hexagonal symmetry of the custom object yields similar facial features equivalent to Face 1 at alignment angles $\theta = 0°$, 60°, 120° and for Face 2 at alignment angles $\theta = 30°$, 90°, 150°. Data was acquired when the object was kept at distance $z = 10$ mm from each CAPSENSAR (L) and CAPSENSAR (R).

The MC carried out a series of identical computational tasks on separate $z$-matrices data obtained for a pair of oppositely aligned CAPSENSAR (L, R) at different $\theta$ to identify the most convenient pair of opposite faces for possible gripping. Since our target object has six lines of symmetry, the macroscopic landscape of the exposed face as obtained for $\theta = 0°$, $60°$, $120°$ are equivalent and designated as Face 1, while that for $\theta = 30°$, $90°$ and $150°$ as Face 2. Thus, the $z$-matrices obtained for Face 1 and Face 2 of the object are representative of faces corresponding to different $\theta$ as described above and are illustrated as image profile with black/white codes as shown in Fig. 3c, d, respectively. The $z$-matrices data for Face 1 and Face 2 are illustrated as 3D $z$-contour representation using $xyz$ gridding of the $z$-matrices as depicted in Fig. 3e, f, respectively. The 3D z-contour representation of a given object face illustrated the landscape of that face and also provided a contactless estimation of its dimension. This estimation of the dimensions of the object face using 3D $z$-contour plot facilitated the determination of the area of the flattest regions on opposite faces of the object. The determination of the area of the flattest regions were obtained by computing the piecewise derivatives of the $z$-matrix elements, corresponding to each $(i, j)$ sensor unit, along the $x$-(BE) and $y$-(TE) directions. The $z$-elements corresponding to the flattest region yielded a null element in the $(5 \times 4)$ $z$-derivative matrix computed along $x$-axis as $\frac{\partial z}{\partial x} = 0$ and $y$-axis as $\frac{\partial z}{\partial y} = 0$. The $z$-derivative matrices obtained separately for both CAPSENSAR (L, R) for each alignment angles $\theta$ were compared with each other to determine the agreeable pair of flattest face of the object using a set of logical commands as described in Table 1.

Since the null element in the derivative matrix signify flat region on the face of the object, the MC recorded the number of null elements from the respective derivative matrices obtained separately from CAPSENSAR (L) and CAPSENSAR (R) at each $\theta$ and denoted as, $N_L$ and $N_R$, respectively. The values $\alpha_\theta = |N_L - N_R|$ and $\beta_\theta = |N_L + N_R|$ for each $\theta$ were computed and stored in MC. After capacitance scanning at six different alignments of the gripper palms arrangements, that $\theta = \theta_{grip}$ corresponding to the minimum value of $\alpha_\theta$ and maximum value of $\beta_\theta$ was identified for subsequent gripping task. The $\theta_{grip} = |Min(\alpha_\theta), Max(\beta_\theta)|$ represented that angle of alignment $\theta$ where the pair of fittest gripping faces with largest flat area was identified for agreeable gripping of the object.

In this work, the object and the pair of CAPSENSARs on robotic palms were aligned along the central axis. The object was placed between the pair of the gripper palms (L, R) on the $x$–$z$ plane and the pair of robotic palms was aligned along the direction of largest dimension of the object at initial $\theta = 0°$ as shown in Fig. 4a. The pair of robotic palm arrangement scanned different object faces through six different $\theta$ to recognize $\theta = \theta_{grip} = 30°$ as the pair of fittest faces with the flattest surface as calculated in Table 1. The MC instructed the rotation servo to align the pair of gripper palm arrangement at this $\theta_{grip}$ as shown in Fig. 4b. At $\theta = \theta_{grip}$, the MC continued to perform the same series of pre-programmed iterative computational tasks to update the $N_L$ and $N_R$ values. Now the grip controlling servo was triggered to initiate the gripping process. The grip controlling servo triggered the movement of gripper palms 1 and 2, bearing CAPSENSAR (L) and CAPSENSAR (R), respectively, which converged on the object to make a gentle contact with $P = 0.5$ kPa (Fig. 4c) on the oppositely situated pair of flattest face to determine $P_{grip}$ required for safe and effective gripping of that particular object. Secure and effective gripping of the test object was performed by grasping the object with calculated $P_{grip}$ to ensure deformation-less and slippage-free operation as shown in Fig. 4d and described below.

**Secure gripping of object.** The cognitive operations for secure gripping of the object were performed when the $\theta = \theta_{grip}$ was identified as shown in Fig. 5a. The MC estimates the area of the flattest surface on opposite faces of the object to determine the $P_{grip}$ and monitor any deformation and slippage incurred on the object. The algorithm (Fig. 2) for the determination of optimum gripping area and hence, execution of intelligent gripping of the test object is described as follows:

During gripping when the CAPSENSAR (R, L) on the gripper palm 1 and 2 were in gentle contact with the pair of gripping faces of the object at $\theta = \theta_{grip} = 30°$, the measurement mode in the CAPSENSARs made a change from the proximity sensing to pressure sensing, which measured the reactionary pressure exerted by the opposite faces of the object on respective devices. Here, the non-linear proximity sensing calibration curve on gentle contact with the object ($z = 0$) transformed into a linear pressure sensing calibration curve with variation in the displacement $\Delta \delta$ of elastomeric Ecoflex dielectric layer of the CAPSENSAR as shown in Fig. 5b. Negative displacement of object towards the CAPSENSAR beyond $z = 0$ mm (as depicted in Fig. 5b) in the pressure sensing regime corresponded to compressive forces (-$z$) on the separate $(i, j)$ pressure sensing units of the CAPSENSAR due to the exerted pressure on the object during gripping (supplementary discussion 2.3). While in contact with the target object at pre-loaded pressure of $P = 0.5$ kPa, the CAPSENSAR (L, R) generated separate $P$-contour plots from their respective recorded capacitive matrices using calibration expression of supplementary Eq. (16) for opposite gripping faces of the object. Fig. 5c shows the $P$-contour plot of one of the opposite gripping faces of the object at $\theta_{grip} = 30°$ under gentle contact. Under this

---

**Table 1 Determination of gripping pressure $P_{grip}$.**

| Target object | Hex nut and bolt arrangement | |
|---|---|---|
| Material of target object | Stainless steel | |
| Weight (m) and dimension of target object | Weight $m = 65$ g and Dimensions shown in Fig. 1c | |
| Initialization for COGBOT operation | Gripper palms aligned parallel to Face 1 considered as $\theta = 0°$ Ref. Fig. 3b and Supplementary Fig. 12 | |
| Alignment angle ($\theta$) | $\theta = 0°$, $60°$, $120°$ (Face 1 of object) | $\theta = 30°$, $90°$, $150°$ (Face 2 of object) |
| Determination of $\alpha_\theta$ and $\beta_\theta$ at different $\theta$ | $\alpha_\theta = |N_L - N_R| = 0$, and $\beta_\theta = |N_L + N_R| = 6$, at $\theta = 0°$, $60°$, $120°$ (Face 1 of object) From Fig. 3c, as $N_L = 3$, $N_R = 3$ | $\alpha_\theta = |N_L - N_R| = 0$, and $\beta_\theta = |N_L + N_R| = 12$, at $\theta = 30°$, $90°$, $150°$ (Face 2 of object) From Fig. 3d, as $N_L = 6$, $N_R = 6$ |
| Determination of $\theta_{grip}$ | $|Min(\alpha_\theta), Max(\beta_\theta)| = |0, 12|$ at $\theta = 30°$, $90°$, $150°$ and thus, $\theta_{grip} = 30°$ | |
| Determination of $P_{grip}$ | $P_{grip} = max|P_{gripL}, P_{gripR}| = 3.6$ kPa, where $|P_{grip}|_{L,R} = 1.2 \times P_{crit}$ and $P_{crit} = 3$ kPa, calculated from Eq. 1 using $\Lambda_{L,R} = N_{L,R} \times \Lambda_{unit} = 6 \times 36 = 216$ mm$^2$, $\Lambda_{unit} = 36$ mm$^2$ | |

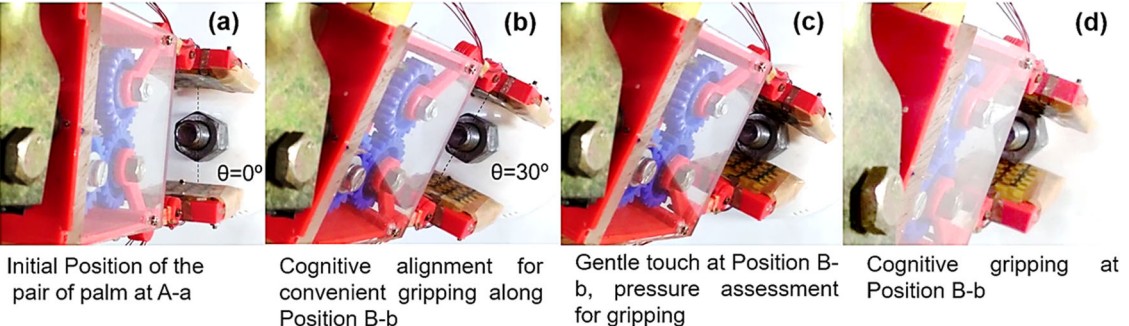

**Fig. 4 Top view images depicting major steps in the cognitive gripping process of COGBOT. a** Initialization steps involving the alignment of the pair of robotic palm arrangement along the direction of widest dimension of the object to angle $\theta = 0°$, **b** Alignment of the pair of robotic palm arrangement at the fittest pair of flattest opposite face of the object at an angle $\theta = 30°$ after capacitive scanning of six different faces at six different $\theta$, **c** Gentle touch with $P = 0.5$ kPa at the fittest pair of flattest opposite faces of the object at an angle $\theta = 30°$ to determine the optimum pressure $P_{grip}$ for gripping, and **d** Cognitive gripping of the object with optimum pressure $P_{grip} = 3.6$ kPa.

condition the iterative cycle facilitated both CAPSENSAR (L, R) to re-estimate $N_L$ and $N_R$ from closest proximity so that the respective area of contact $\Lambda_L$ and $\Lambda_R$ on opposite gripping faces of the object could be accurately determined. Since the area of each pressure sensing unit, i.e., $\Lambda_{unit} = 36$ mm$^2$, the flattest areas of contact for gripping, as obtained for respective CAPSENSAR (L, R), were determined to be $\Lambda_L = N_L \times \Lambda_{unit}$ and $\Lambda_R = N_R \times \Lambda_{unit}$, respectively, and stored in the MC. The critical gripping pressure $P_{crit}$ required for deformation and slippage-free gripping was determined to be

$$P_{crit} = \frac{mg}{Min(\Lambda_L, \Lambda_R)} \quad (1)$$

where $m$ and $g$ denote the mass of the object and gravitational acceleration, respectively. The value of $P_{crit}$ is calculated in MC and thereafter instructs the grip controlling servo motor to grip the object with a gripping pressure of $P_{grip} = 1.2\ P_{crit}$, providing an additional tolerance pressure of $0.2\ P_{crit}$ to ensure successful and reliable grip. In this work using the custom-made object $P_{grip}$ was graphically calculated from supplementary Fig. 13a, b to be 3.6 kPa as summarized in Table 1. This value of $P_{grip} = 3.6$ kPa was confirmed from the $P$-contour representation when the object was gripped with optimal pressure $P_{grip}$ as shown in Fig. 5d. Since the gripper is able to perform the gripping action with measured $P_{grip}$, it can be used for the gripping of the delicate object. The gripping of the object with $P_{grip}$ enables slippage-free and secure operation as confirmed by stable orientation of the object between the pair of gripper palms and illustrated in Fig. 5e. Secure gripping of the object was monitored by comparing the $P$-matrices and the $P$-contour plots before and during gripping as shown in supplementary Fig 13d, e.

**Anti-slippage algorithm.** The anti-slippage algorithm prevents undesirable slippage and deformation and ensures safe gripping of the object during the operation of the COGBOT. This may arise when the pair of CAPSENSAR integrated palms of the COGBOT were not parallelly aligned or were unaligned due to various factors including overloading. The anti-slippage algorithm was used to detect variation in physical movement and deformation of the object during gripped condition as described in Supplementary discussion 6. For secure griping of the object, the $P_{grip}$ exerted by the CAPSENSAR (L, R) at the appropriate faces of the object in the gripped condition was monitored using the $P$-matrices plots obtained from each of the devices (L, R). Any slippage of object, if detected by the $P$-matrices plots was prevented using an iterative process, which revised the applied $P_{grip}$

on the object to resist slippage. This anti-slippage cognitive operation by COGBOT is described as follows:

The CAPSENSAR (L) and CAPSENSAR (R) continuously acquired the data in the gripped state and subsequently generated the $P$ matrices plots for each iterative cycle. Each iterative cycle yields new pattern and number ($N_L$ and $N_R$) of null elements in the $P$-matrices. Since the iterative cycles were continuously updated throughout the entire duration of operation, the Min $|\Lambda_L, \Lambda_R|$ and hence the values of $P_{crit}$ and $P_{grip}$ were revised in the MC accordingly. The slippage of a solid object grasped between the palms of the COGBOT was detected by observing the translational shift in null elements across the $P$-contour matrix. The deformation of the grasped object was detected by any relative variation in the area $\Lambda_L, \Lambda_R$. Since our custom-made object was solid stainless steel, the initial slippage was detected by its translational motion under gravity. Figure 5f, g shows the pressure landscapes obtained from CAPSENSAR (L) and (R), respectively, in the subsequent iterative cycle during undesirable slippage of the target object as depicted in Fig. 5h. Under post slippage circumstances, the new $N_L$ and $N_R$ and hence the new area of contact $\Lambda_L^{new}$ and $\Lambda_R^{new}$ were re-estimated from the $P$-matrices representation of CAPSENSAR (L) and CAPSENSAR (R), respectively, in subsequent iterative cycle. Since the null element reduced due to slippage induced translational movement of the object, the newly obtained gripping pressure $P_{grip}^{new}(>P_{grip}) = \frac{mg}{Min(\Lambda_L^{new}, \Lambda_L^{new})} = 4.3$ kPa was used to grasp the object to resist slippage (Supplementary Fig. 14 and Supplementary Table 2). The iterative process for anti-slippage operation was continued till no further slippage of the object was detected. Thus, the COGBOT using the CAPSENSARs is capable of detecting local deformation or slippage of the target object in real time, ensuring reliable gripping of delicate objects. The COGBOT is capable of exerting a maximum $P_{grip} = 5$ kPa (Supplementary Fig. 15) and are suitable for soft and light-weight object. The grip performance may be improved by suitable redesign of the end-arm tool.

**Conclusion**
This work reports the development of printed graphene ink based capacitive multi-sensing array (CAPSENSAR) for proximity and pressure sensing. The CAPSENSAR operates on the variation of fringing field capacitance where the top and the bottom electrode array of the device were designed with optimized non-overlapping area between them to achieve high sensitivity of 0.012 pF mm$^{-1}$ and wide dynamic range of 0–120 mm. The proximity sensor array of the CAPSENSAR generates contactless proportion sized-capacitive impression of exposed face of the

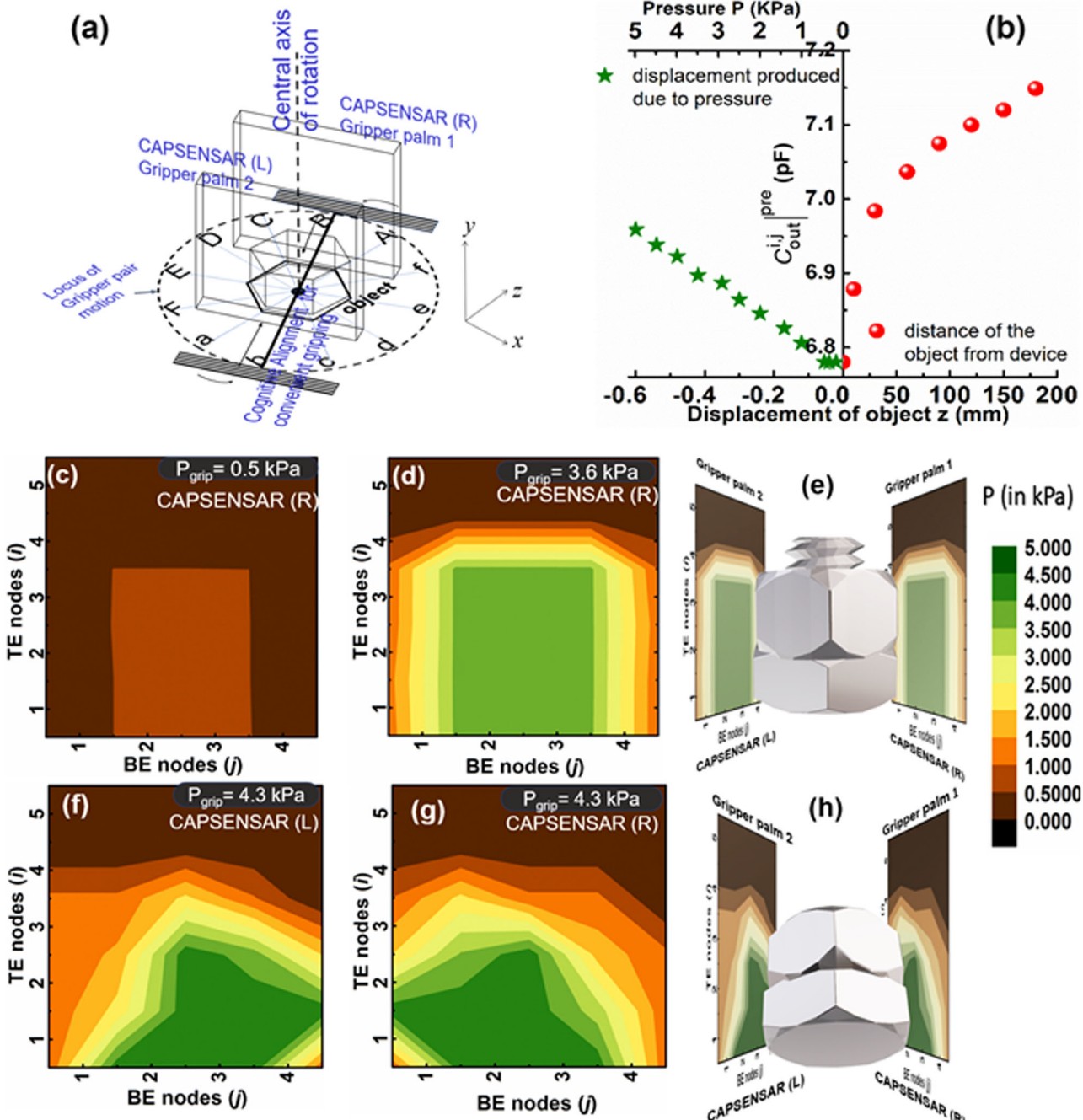

**Fig. 5 Secure gripping of object and anti-slippage operations. a** Schematic illustration of the identified gripping face showing the alignment angle convenient for gripping, **b** variation of output capacitance $C_{out}^{i,j}]^{pre}$ with the displacement z of the object before and after its contact on the CAPSENSAR showing the transition in the measurement of $C_{out}^{i,j}]^{pre}$ between the proximity and the pressure sensing regimes with a dead band region in the range 0.5–0 kPa. *P*-contour representation for the fittest pair of opposite flattest faces at $\theta = 30°$ for **c** gentle touch with $P = 0.5$ kPa, showing the flattest area of the face and **d** reliable gripping with $P_{grip} = 3.6$ kPa, showing the gripping area and **e** orientation of the object during secure gripping. *P*-contour plot obtained from **f** CAPSENSAR (L) and **g** CAPSENSAR (R) showing undesirable slippage of the object leading to the revision of the $P_{grip}$ to its new value of $P_{grip} = 4.3$ kPa, **h** changed orientation of the test object during slippage.

object and subsequently, constructs its corresponding three-dimensional landscape illustration using graphical contour representation. This feature of the CAPSENSAR was computationally utilized in the detection of the fittest pair of flattest opposite faces of the given object and employed for convenient, effective and stable gripping of the object using the proposed cognitive robotic gripper (COGBOT). The proximity sensor offers good *z*-resolution of 0.09 mm within $z = 3$ mm thereby providing good depth accuracy in *z*-contour gridded matrices. An

excellent response time of 0.3 s makes the sensor suitable for fast capacitive scanning of the objects. The pressure sensor array of the CAPSENSAR generates capacitive impression of the gripped face of the object in terms of pressure contour representation, which was utilized to ensure slippage-free and deformation-free gripping of the object through cognitive computations. A pair of CAPSENSARs was integrated on the palms of the robotic gripper to set-up the COGBOT. The COGBOT is capable of controlled and precise gripping of target object by simple pair of

end-arm-tool using an algorithm with a series of cognitive decision-making steps-detection of face landscape, determination of fittest pair of opposite gripping faces, measurement of suitable gripping force, continuous monitoring of gripping force to prevent damage and slippage while gripping. The cognitive gripping strategy by COGBOT makes the robotic gripper free of mechanical complexities in terms of end-arm-tool architecture and movable joints, yet efficient in performance. The COGBOT utilizes capacitive principles for proximity and pressure sensing in CAPSENSAR, and thus simplifies the electronic circuitry, which provide feedback signal to controlling unit. This makes the computational programming easy, leading to the fast implementation of cognitive processes for effective gripping. The CAPSENSAR utilizes cost-effective materials, follows a mask-less fabrication technique with customizable electrode design, avoids wastage of printable ink due to printed electrodes and circuitry, to makes it commercially viable for rapid production and cost-effective manufacturing. Since the device has an overall thickness of 1.03 mm it can be easily conformable on end-arm tools of robotic grippers, thereby facilitating reliable gripping. The use of biocompatible materials such as Ecoflex, PVA, PI and graphene in the CAPSENSAR makes the end-arm tool of the COGBOT eco-friendly for use in household domestic task, healthcare service for automated robotic delivery, robot aided catering services and bionic arms. However, in spite of various advantages of the COGBOT, it suffers from numerous shortcoming that must be addressed for effective usage. Although the proximity sensor array of the CAPSENSAR offers a high $z$-axis resolution of 0.091 mm for $z < 30$ mm, it suffers from low $x-y$ flatness resolution of 3 mm and further deteriorates with $z$. Thus, the CAPSENSAR may generate a distorted and blurred capacitive impression of the object when kept at large distances $z > 60$ mm from the device. The $x-y$ flatness resolution of the CAPSENSAR was limited by the elementary sensor unit dimension $a = 3$ mm and the interelectrode spacing $\zeta = 3$ mm and thus features on the object with dimensions <3 mm may not be clearly distinguished during shape detection by the device unless the features are sharp enough to generate high surface charge density. Although the proposed CAPSENSAR was bendable, the estimation of object shape using a curved device may not yield the right shape of the target object unless the device was recalibrated incorporating the curvature of the device during measurement. During determination of the fittest pair of opposite face of the object prior to gripping, the palm must be parallelly aligned and mutually facing each other to facilitate accurate determination of the $P_{grip}$ to ensure safe gripping. Any deviation in alignment may lead to flawed $P_{grip}$ resulting in slippage. To utilize the motion detection ability of the CAPSENSAR, the future work on the COGBOT will includes working with moving objects. For this purpose, the response time of the detection and the time required for the execution of the iterative cycle in the program must be improved.

## Methods

The device was fabricated in six steps(a-h) (Supplementary Discussion 3): (a) The polyimide (PI) sheet was properly cleaned and made hydrophilic by using the O$^{2-}$ plasma to achieve the good printability for graphene ink of viscosity=8 mPa.s. Then this low viscous printable graphene ink was ink-jet printed on the processed surface of PI to realize the 200 nm thick top electrodes (TEs) as shown in Supplementary Fig. 7a-Inset1. The 300 Scm$^{-1}$ high conductivity of the TEs was achieved by heating the printed graphene layer in the vacuum oven at a temperature ($T$) of 250 °C for 30 min. Supplementary Fig. 7a-Inset 2 shows that the non-contact ink-jet printing method (using print head of diameter=30 µm) was mask-less and material wastage free and is fast with print throughput of 15 min and printing rate 20 mm/s.(b) The TEs were masked by spray coating the Polyvinyl alcohol (PVA) followed by its curing at $T = 60$ °C. (c) The PI sheet is flipped to make its reverse side exposed for bottom electrodes (BE) printing. Then, this exposed PI side is prepared for BE printing by making it clean and hydrophilic in the similar method as discussed in step (a). (d) The graphene-based BEs were ink-

jet printed on the processed surface of the reverse side of PI in similar technique of step (a) and dried in vacuum oven at a low temperature of 60 °C. (e) Then the PVA masking layer on the TE was removed by rinsing it using DI water followed by the heating of BEs in the vacuum oven at a temperature ($T$) of 250 °C for 30 min to achieve their high conductivity as discussed in step (a). (f) After this step, the rear side cladding of the device was realized by spin coating the ecoflex solution on the BE at a rotation speed of 500 rms and subsequently, cured it at $T = 60$ °C. (g) The device is again flipped to passivate its top electrodes by spray coating the PVA layer and subsequently, curing it at $T = 60$ °C. The image of the fabricated CAPSENSAR and its integration on the gripper palm are shown in Supplementary Fig. 7h and Supplementary Fig. 7h-Inset 3, respectively. The CAPSENSAR utilizes cost-effective materials, follows a mask-less fabrication technique with customizable electrode design, avoids wastage of printable ink due to printed electrodes and circuitry, involves easy fabrication steps-ink-jet printing of graphene electrodes on PI, spin coating of Ecoflex on BE and spray coating of PVA on TE and fast printing with print throughput of 15 min and printing rate 20 mm s$^{-1}$, makes it commercially viable for rapid production and cost-effective manufacturing.

## Data availability

The data generated and analyzed during the study are available from the corresponding authors on a reasonable request.

## Code availability

The code used during the study are available from the corresponding authors on a reasonable request.

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

## Acknowledgements

The authors acknowledged funding from Department of Science and Technology (DST) and Science and Engineering Board (SERB) (Grant No. SPF/2021/000021 and IMP/2019/000237).

## Author contributions

T.M. conceived the idea, designed the project, conducted the fabrication, characterization and data analysis, conceptualized the robotic operations, performed the demonstration activities, and wrote the manuscript. D.G. conceptualized of the project, acquired the funds, administered the project, and reviewed the manuscript.

## Competing interests

The authors declare no competing interests.
