## [Peer Review File · Communications Engineering]

Reviewers' comments:

Reviewer #1 (Remarks to the Author):

This work reports a fully printed capacitive multi-sensor array for cognitive decision-making operations of cognitive robotic gripper. The sensor array can sensor both proximity and pressure. For proximity sensing, the sensor array can generate the contactless-capacitive impression of the object to construct a proportion-sized three-dimensional landscape of the exposed face of the object. On the other hand, the pressure sensor array can generate the impression of the gripped object face at the computationally determined optimum gripping pressure. There are some points for the authors to consider:

1. 'KPa' should be 'kPa'.
2. Figures 8(b) and 9(b), as the capacitance is proportional to the size of capacitive sensor, the capacitance changes (ΔC) as a function of distance/pressure should be normalized as their initial capacitance (C_0) to get rid of the size effect.
3. As for the proximity sensing, what is the target object? Does the capacitance change remain the same if it is changed to another object?
4. Why there is a dead band region in Figure 9(b)? Capacitive pressure sensors are usually more sensitive in the low pressure range and then saturate in the high pressure range.
5. The working pressure range of the graphene printed electrodes can be determined by monitoring the resistance changes of the electrodes under different pressure.
6. Are there any results to support the detection of slippage?

Reviewer #2 (Remarks to the Author):

The authors report cognitive gripping with graphene printed multi-sensor array and presented the work well. The minor comments to the authors are as follows.

1. Do the coplanar electrodes' size and the gap between them optimized for evaluating the sensors' performances?
2. What is the average resolution that can be achieved by the ink-jet printer?
3. Is the graphene ink commercially purchased? If not, it is recommended to show the chemical composition of the ink-jet printed films to confirm the presence of graphene.
4. Can the conductivity of the ink-jet printed films be increased? Because high conductive inks are available commercially.
5. What is the advantage of graphene ink compared to other conductive ink such as Ag?
6. Restrict the Figures and tables to 10 in the manuscript file per the journal norms, and the remaining figures are suggested to be inserted in the supplementary information.

Reviewer #3 (Remarks to the Author):

This study reports a printed graphene ink-based capacitive multi-sensing array (CAPSENSAR) for

proximity and pressure sensing based on the variation of fringing field capacitance. It is an interesting study that combines object shape sensing, surface morphology detection, pressure adaptation, and feedback. The authors collected a plethora of data. However, some critical questions were not answered yet.

1. literature review is not complete. Current advances by other researchers and their drawbacks were not discussed, including the shape/size/morphology of cognitive sensors and pressure adaption grippers. The authors focused only on their own advantages and overlooked their disadvantages in the conclusion and future work.
2. What is the response time of the authors' prototype? What is the influence of the array size (i,j)? 6 mm and 21 mm are quite small sizes in domestic applications. Could the authors justify why such dimensions were used? is this because of lab restriction or some other reasons?
3. The authors showed significant differences in responses to distinct shapes of objects. What are the resolutions of flatness and surface morphology of the prototype?
4. How do the object materials (e.g., electric/dielectric properties) impact the capacitance and, therefore, the performance of the palm? They should be considered in the theoretical analysis, simulation, and experiments.
5. The theory part is weak and considered only a sphere. It should be more comprehensive and convincing to support the simulation and experiments.
6. The comparison among theory, simulation, and experiments should be added. Why do equation 6 and equation 5 have different forms of relations between capacitance and distance?
7. The quality of the presentation could be improved. In Figure 2, a schematic is needed to make clear the structure. It is hard to read now. In Figure 8, the capacitance changes over time. A detailed explanation should be added to the caption, explaining the operation at the changing time points. The manuscript could be more precise and focused, highlighting the innovation of this study.

Reply to reviewers:

Reviewer #1 (Remarks to the Author):

This work reports a fully printed capacitive multi-sensor array for cognitive decision-making operations of cognitive robotic gripper. The sensor array can sensor both proximity and pressure. For proximity sensing, the sensor array can generate the contactless-capacitive impression of the object to construct a proportion-sized three-dimensional landscape of the exposed face of the object. On the other hand, the pressure sensor array can generate the impression of the gripped object face at the computationally determined optimum gripping pressure. There are some points for the authors to consider:

1. 'KPa' should be 'kPa'

The authors thank the reviewers for pointing out this.

This abbreviated unit is modified in the **Fig. 2 (e) and (f) of page 8, 4(c) and (d) of page 13 and 9 (a-ii), (b) and (c) of page 22** and their respective captions in revised manuscript. The KPa is also replaced by kPa in the caption of **Fig. 8(c) of page 20** of revised manuscript.

The KPa is modified as kPa in the text of **page 11-paragraph 1, page 15-paragraph 3, page 16-paragraph 1,2, page 23-paragraph 3, 4 and page 24-paragraph 1** of revised manuscript.

2. *Figures 8(b) and 9(b), as the capacitance is proportional to the size of capacitive sensor, the capacitance changes (ΔC) as a function of distance/pressure should be normalized as their initial capacitance (C_0) to get rid of the size effect.*

The authors thank the reviewers for the suggestion. The normalized change in output capacitance as expressed by $\Delta C_{PROX} / C_{out}^{i,j}]^{abs}$ and $\Delta C_{PRESS} / C_{out}^{i,j}]_{z \rightarrow 0}^{pre}$ are plotted against $z(\text{mm})$ and $P(\text{kPa})$ as shown in **Fig. 4(b) and 4(d) of page 13** of revised manuscript respectively. The **Fig. 8(b) (page 13 of old manuscript)** and **Fig. 9(b) (page 15 of old manuscript)** are re numbered as **Fig. 4(b)** and **Fig. 4(d) (page 13 of revised manuscript)** respectively in the modified manuscript and the Fig. 4 appear as:

Figure 4: (a) Dynamic measurement of output capacitance $C_{out}^{i,j}$ [pF] with time for an approaching object at different distances $z = 0, 5, 10, 30, 60, 90, 120,$ and 150 mm, recorded by the (i, j) sensor unit of CAPSENSAR (b) Normalised change in output capacitance $\Delta C_{PROX} / C_{out}^{i,j}$ [abs] vs. proximity distance z calibration curve of a (i, j) sensor unit for an object of stainless steel, relative to the results obtained from theoretical and simulation studies (c) Dynamic measurement of output capacitance $C_{out}^{i,j}$ [pre] with time for different applied pressures $P = 0.1, 0.5, 1, 2, 3, 4$ and 5 kPa, showing alternative pressure and release cycles when the object was approached onto the (i, j) sensor unit from a distance of $z = 180$ mm for each cycle (d) Normalised change in output capacitance $\Delta C_{PRESS} / C_{out}^{i,j}$ [pre] vs. Pressure P calibration curve of (i, j) sensor unit showing a dead band in the range $0-0.5$ kPa and its comparison with simulation results

3. As for the proximity sensing, what is the target object? Does the capacitance change remain the same if it is changed to another object?

The authors thank the reviewers for their suggestion. A custom-made steel object is taken as the target material.

Since the charge density on the surface of a metallic object (conductor) with finite conductivity depends on the skin depth of the metal at a fixed frequency, the surface charge density is dependent on the metallic property (conductivity) of the object. As the conductivity of the metallic object material increases the skin depth of the charged conductor decreases, thereby increasing the surface charge density on the object, which increases the change in output capacitance of the device. Thus, the output capacitance of the sensor unit decreases in presence of metallic object and this change is high when the conductivity is high. Thus, the change in output capacitance varies when the experiment is performed with object of different metal having same shape and size.

However, when an object of dielectric material is introduced in the proximity of the sensor unit the output capacitance increases with the relative dielectric permittivity of the object of the same shape and size.

The effect of the device output capacitance on the object material, size and shape are newly included in *Sec. 3.1-page 6-paragraph 4* of revised manuscript and appears in revised manuscript as:

The ΔC_{PROX} increases sharply when the object approaches $z \rightarrow 0$, while the function decays to zero at large $z \rightarrow \infty$ to yield $C_{out}^{i,j}]^{pre} \rightarrow C_{out}^{i,j}]^{abs}$. Since the charge density on the surface of a metallic object (conductor) with finite conductivity depends on the skin depth of the metal at a fixed frequency, the σ_{obj} is dependent on the metallic property of the object. Thus the ΔC_{PROX} varies when the experiment is performed with object of different metal having same shape and size. Although the σ_{obj} is a shape dependent quantity^{38, 39}, the ΔC_{PROX} of the single sensor unit is affected when the dimension of the local curvature $\ll a=3$ mm i.e sharp. Thus for objects of dimensions $\geq a$, exposed to the sensor unit at a fixed z , may be considered as planar for which ΔC_{PROX} is independent of shape and size of the object and solely depends on z . However, for an array of sensor units, each sensor unit records their respective ΔC_{PROX} to construct a capacitive landscape of the segmented face of the object. The dimensions and surface morphology variations of the object can be computed from the calibration curve of the proximity sensor array. Thus, metallic objects (with dimensions 30×24 mm) of various shapes and sizes can be distinguished from capacitive impressions generated by the CAPSENSAR. On the contrary, when a dielectric material of permittivity ϵ_r is introduced at a proximal distance z from the (i, j) sensor unit, the $C_{out}^{i,j}]^{pre} > C_{out}^{i,j}]^{abs}$ as the effective dielectric thickness of the device decreases (see Supp. Sheet Sec.1.2).

4. Why there is a dead band region in Figure 9(b)? Capacitive pressure sensors are usually more sensitive in the low-pressure range and then saturate in the high-pressure range.

The authors are pleased to answer the reviewer's query. The pressure sensor shows a dead band in the range 0-0.5 kPa. This is attributed to the local stiffness of the 30 μ m thick PI sheet bearing the printed electrodes, constituting the proximity sensor array in the device. This stiffness of the PI sheet is unable to produce measurable change in output capacitance $C_{out}^{ij}]^{pre}$ and thus limits the detection of very low applied pressure in the range 0-0.5 kPa termed as the dead band region. Due to the presence of the dead band region in the pressure sensor, the estimation of appropriate gripping force P_{grip} at the pair of flattest face of the object using the proximity sensor array can be effectively measured with high accuracy. The role of the proximity sensor array in the estimation of P_{grip} is discussed elaborately in **Sec. 8.2-page-18-paragraph-4** and **Sec. 8.3-page-23-paragraph-2** of revised manuscript. The optimum gripping force was measured using the relation (10) (**page 23** of revised manuscript) as:

$$P_{grip}=1.2 \times \frac{mg}{\text{Min}(\Lambda_L, \Lambda_R)}, \text{ where } \Lambda_{L,R} = N_{L,R} \times \Lambda_{unit} \text{ and } N_L \text{ and } N_R \text{ represents number of null}$$

elements from the respective derivative matrices obtained separately from CAPSENSAR (L) and CAPSENSAR (R) respectively at each θ .

The text on dead band region in **Fig. 4(d) of page 13** of revised manuscript (or **Fig. 9(b) of page 15** of old manuscript) is newly added in **Sec. 7.1.2-page 16-paragraph 1** of the revised manuscript and appears in the manuscript as:

The (i, j) pressure sensor units suffer a dead band region between 0-0.5 kPa where the sensor unit showed no variation in $\Delta C_{PRES} / C_{out}^{i,j}]_{z \rightarrow 0}^{pre}$. This is attributed to the local stiffness of the 30 μ m thick PI sheet bearing the printed electrodes, constituting the proximity sensor array in the device. This stiffness of the PI sheet is unable to produce measurable change in output capacitance $C_{out}^{ij}]^{pre}$ and thus limits the detection of very low applied pressure in the range 0-0.5 kPa termed as the dead band region.

5. The working pressure range of the graphene printed electrodes can be determined by monitoring the resistance changes of the electrodes under different pressure.

The authors are pleased to answer the reviewer's query. The capacitive sensors are designed with separate arrays of top and bottom electrodes which may not be suitable for resistive investigations. However resistive analysis may be performed under bend condition when measurements are performed across the graphene electrodes in either plane of the device. Under applied pressure on the device leading to bending of electrodes, the variation in resistance may be attributed to the shearing of graphene micro flakes over each other in graphene printed electrode. However, in our case, since the device was fixed on a rigid support where there is limited scope for electrode bending, the measurements acquired between neighboring sub-electrodes on PI sheet yielded no significant change in resistance under applied pressure. Freely bendable device may produce change in resistance but will damage the 300 nm thick graphene printed electrodes under repetitive use, thereby hindering long term usability.

6. Are there any results to support the detection of slippage?

The authors thank the reviewers for their suggestion. The results for the detection of slippage that were acquired during investigation are presented here. The slippage occurred due to non-alignment of gripper jaws which led to improper grasping and consequent dis-orientation of the target object under gripped state. Detailed discussion on the prevention of slippage is provide in **Sec. 3-page-7-paragraph 2 and Fig. S5(c-d)** of Supp. Sheet.

The experimental data illustrating the COGBOT capability to resist slippage is newly added in the **Sec. 8.4-page 23-paragraph 5** of the revised manuscript and appears in the manuscript as:

The anti-slippage algorithm prevents undesirable slippage and deformation and ensures safe gripping of the target object during the operation of the COGBOT. Such case may arise when the pair of CAPSENSAR integrated on the palms of the COGBOT were not parallelly aligned or were unaligned due to various factors including overloading. The anti-slippage algorithm was used to detect variation in physical movement and deformation of the object during gripped condition. The COGBOT exerted a pressure of P_{grip} to grip the object and was confirmed by each (i, j) pressure sensing unit of the CAPSENSAR attached to the gripper palms. Since the iterative cycle for the determination of N_L and N_R was continuously updated throughout the entire duration of operation, the $\text{Min} |\Lambda_L, \Lambda_R|$ and hence the values of P_{crit} and P_{grip} were updated in the MC accordingly as illustrated in Supp. Sheet Sec. 3. Fig. 9(c-i) and (c-ii) shows the pressure landscapes of CAPSENSAR (L) and (R) respectively in the subsequent iterative cycle during undesirable slippage of the target object. The deformation of the grasped object was detected by any relative variation in the area Λ_L, Λ_R and the planar distribution of the null elements in the P-contour matrix signified deformation of the object. The slippage of a solid object grasped between the palms of the COGBOT was detected by observing the translational shift in null elements across the P-contour matrix. Since our custom-made object was solid stainless steel, the initial slippage was detected by its translational motion under gravity. Under post slippage circumstances, the new N_L and N_R and hence the new area of contact Λ_L^{new} and Λ_R^{new} were re-estimated from the z-matrices representation (See. Supp Sheet Sec. 3) of CAPSENSAR (L) and CAPSENSAR (R) respectively in subsequent iterative cycle. Since the null element reduced due to slippage induced translational movement of the object, the newly obtained gripping pressure $P_{grip}^{new} (> P_{grip}) = \frac{mg}{\text{Min}(\Lambda_L^{new}, \Lambda_L^{new})} = 4.3 \text{ kPa}$ was used to grasp the object to resist slippage.

Thus, the COGBOT using the CAPSENSARs is capable of detecting local deformation or slippage of the target object in real time, ensuring reliable gripping of delicate objects.

Figure 9: (b) P -contour representation for the fittest pair of opposite flattest faces at $\theta=30^\circ$ for (i) gentle touch with $P=0.5$ kPa, showing the flattest area of the face and (ii) reliable gripping with $P_{grip}=3.6$ kPa, showing the gripping area. (c) P -contour plot illustrating the prevention of slippage (associated with decrease in N_L and N_R) by the revision of the P_{grip} to its new value of $P_{grip}=4.3$ kPa.

Reviewer #2 (Remarks to the Author):

The authors report cognitive gripping with graphene printed multi-sensor array and presented the work well. The minor comments to the authors are as follows.

1. Do the coplanar electrodes' size and the gap between them optimized for evaluating the sensors' performances?

The authors are pleased to provide an explanation to this query. For the output capacitance of the sensor unit of CAPSENSAR to be in the measurable range ~tens of pF, the architecture of the device and dimensions of the printed graphene electrodes are optimized to serve the purpose. The increase in the electrode dimension $\sim a$ increases the dynamic range as well as the sensitivity of the sensor unit. However, decreasing the electrode dimension $a < 3$ mm lowers the output capacitance reading of sensor unit below the measurable range ~100 of fF, thereby reducing the dynamic range $z < 10$ mm and the z -resolution ~ 10 mm beyond $z > 20$ mm. The interelectrode spacing ζ increases the uniformity of the fringing field at high z facilitating stable sensitive and consistent detection but reduces the x - y flatness resolution of the sensor unit. Due to the reduction in the x - y resolution, large deviation may be observed during shape detection of object using proximity sensor array with $\zeta > 4$ mm which can lead to distorted capacitive impression of the target object. Thus, devices must be designed with sensor unit dimension $a \geq 3$ mm and inter electrodes spacing $a \leq \zeta \leq (a+1)$ to achieve enhanced performance in shape detection.

This influence of the coplanar electrodes' size and their interelectrode spacing on the device performance is included in Sec. 7.2 - page 18- paragraph 1 and Sec. 9 - page 25- paragraph 1 of the revised manuscript.

The text appears in Sec. 7.2 - page 18- paragraph 1 of revised manuscript as:

The x - y flatness resolution depends on the dimension of electrode size and the interelectrode distance $a, \zeta = 3$ mm. Since the accuracy in the x - y measurement was 80%, the flatness resolution \mathcal{R} at a distance z from the device is $3+(0.2 z)$. The flatness resolution of the device deteriorates as the object was placed further away from the device and was attributed to the diverging fringing electric field of the device.

The text appears in *Sec. 9 - page 25- paragraph 1* of the revised manuscript as:

Although the proximity sensor array of the CAPSENSAR offers a high z -axis resolution of 0.091 mm for $z < 30$ mm, it suffers from low x - y flatness resolution of 3 mm and further deteriorates with z . Thus the CAPSENSAR may generate a distorted and blurred capacitive impression of the object when kept at large distances $z > 60$ mm from the device. The x - y flatness resolution of the CAPSENSAR was limited by the elementary sensor unit dimension $a = 3$ mm and the interelectrode spacing $\zeta = 3$ mm and thus features on the object with dimensions < 3 mm may not be clearly distinguished during shape detection by the device unless the features are sharp enough to generate high surface charge density

2. *What is the average resolution that can be achieved by the ink-jet printer?*

The authors are pleased to answer the reviewer's query. With the use of print head diameter of 30 μm the average resolution achieved between two adjacent printed lines is 300 μm when the width of an ink-jet printed line is 80 μm . The dimension of the print head used for ink-jet printing the graphene electrode of width 3 mm is newly added in *Sec. 6.1-page 11-paragraph 3* of the revised manuscript.

3. *Is the graphene ink commercially purchased? If not, it is recommended to show the chemical composition of the ink-jet printed films to confirm the presence of graphene.*

The authors are pleased to answer this query. The graphene ink was commercially purchased from Sigma Aldrich (793663-5ML). The details of all materials and equipment used for fabrication of CAPSENSAR and COGBOT are added in *Sec. 5- page-11-paragraph 2* of the revised manuscript and appears as:

High temperature heat resistant polyimide (PI) tape (Kapton Tape) for Transfer Printers of thickness 30 μm and Size (24 mm \times 30 m), Smooth Ecoflex 00-30 Soft Silicone Liquid Rubber and 3D Printing Material ABS 3D Printing Filament (Red) with Print temperature - 220-230°C were purchased from Amazon. Graphene ink (793663-5ML) of resistivity 0.003-0.008 Ω cm and Poly (vinyl alcohol) (341584-25G) of Mw 89,000-98,000, 99% hydrolyzed were purchased from Sigma Aldrich, USA. 3D printing of models were implemented using Ultimaker 2+Connect. Graphene printing of electrodes were performed using Jetlab® 4 - Tabletop Printing Platform, MicroFab Technologies, Inc., USA

4. *Can the conductivity of the ink-jet printed films be increased? Because high conductive inks are available commercially.*

The authors are pleased to answer this question. The inkjet printable graphene inks have lower viscosity in the range 7-15 mPa.s relative to screen printable inks and other graphene dispersions available commercially. This commercial product (Sigma Aldrich 793663-5ML) used in the study has the lowest resistivity of 0.003-0.008 Ω cm as compared to other inkjet printable inks. However, the conductivity ($=1/\text{resistivity}$) of the inkjet printed graphene line can be further enhanced by the use of the following methods:

- (a) post printing treatments including thermal annealing and compression rolling of this printed electrode can increase the conductivity of the graphene ink to 4×10^4 S/m [1]
[1] *ACS Appl. Mater. Interfaces* 2019, 11, 35, 32225–32234
- (b) graphene ink prepared by functionalization of graphene flakes with Ag/ Au/Cu nano particles or nanowires provides a conductivity in the order of 10^3 - 10^4 S/m. [2]

[2] Buga, Cláudia, Viana, Júlio. "Inkjet Printing of Functional Inks for Smart Products". *Production Engineering [Working Title]*, edited by Majid Tolouei-Rad, Pengzhong Li, Liang Luo, IntechOpen, 2022. 10.5772/intechopen.104529.

5. *What is the advantage of graphene ink compared to other conductive ink such as Ag?*

The authors are pleased to answer this query. The motivation behind the use of graphene ink lies in the development of bendable electrodes of custom architecture on both sides of the PI sheet and aimed for diverse applications including robotic gripping. The graphene ink-based electrodes are most suitable for bending applications where the graphene micro-flakes in the electrode shears over each other under local strain due to bending and restores back on withdrawal. Moreover, the graphene lattice can withstand a localized strain of upto 20% without suffering damage and hence provides excellent bendability to the graphene based printed electrodes. The Ag based electrodes may develop microcracks upon repetitive use and can result in the shift in baseline of the sensor. Moreover, unlike Ag based electrodes the graphene-based electrodes do not suffer thermal and environmental degradation. [3]

[3] Hwang, Y., Choi, J., Kim, JW. et al. Ag-fiber/graphene hybrid electrodes for highly flexible and transparent optoelectronic devices. *Sci Rep* **10**, 5117 (2020).

The advantage of graphene-based electrode is newly included in **Sec. 1-page 3-paragraph 1** of the revised manuscript and appears in the manuscript as:

The graphene based printed electrodes are most suitable for bendable devices as the graphene micro-flakes in the electrode shears over each other under local strain due to bending and restores back on withdrawal. Moreover, the graphene lattice can withstand a localized strain of upto 20% without suffering damage and hence provides excellent bendability to the graphene based printed electrodes.

6. *Restrict the Figures and tables to 10 in the manuscript file per the journal norms, and the remaining figures are suggested to be inserted in the supplementary information.*

The authors thank the reviewers for their suggestion. The Figures and tables are rearranged to include the important data in the manuscript so that the research conducted can be expressed vividly in the article and the added information are shifted to the supplementary sheet. The new manuscript has 9 Figures and 2 tables.

Reviewer #3 (Remarks to the Author):

This study reports a printed graphene ink-based capacitive multi-sensing array (CAPSENSAR) for proximity and pressure sensing based on the variation of fringing field capacitance. It is an interesting study that combines object shape sensing, surface morphology detection, pressure adaptation, and feedback. The authors collected a plethora of data. However, some critical questions were not answered yet.

1. *literature review is not complete. Current advances by other researchers and their drawbacks were not discussed, including the shape/size/morphology of cognitive sensors and pressure adaption grippers. The authors focused only on their own advantages and overlooked their disadvantages in the conclusion and future work.*

The authors thank the reviewers for the suggestions.

The recent technological advancements in automated robots, their shortcomings and their addressal through the proposed COGBOT are newly included in the manuscript. The following text appear in *Sec. 1- page 1- paragraph 4 and Sec. 1-page 2-paragraph 1* of the revised manuscript:

To diversify robotic applications in the household, at workplace and as medical assist, the robots must be reliable in use, portable, cost effective, low powered and user friendly. Kozłowski et al.²¹ developed robotic compliant jaw gripper with three fingers that execute picking up and drop operations, manipulation of objects at the fingertips and for conformally grasping large and irregular objects between the three fingers. Each finger consists of plurality of interconnected phalanges configured to grasp an object. Due to the multiple end arm tool and numerous phalanges at each finger required for conformal grasping, this gripper possesses complex mechanical design and includes multiple sensors integrated at the jaws which makes the gripper expensive, non-user friendly and unsuitable for domestic and lightweight economic operations. Soft grippers are increasingly investigated for the design of lighter, simpler, and more universal grippers. Guo et al.⁸ developed soft two fingered pneumatic grippers with integrated stretchable electro-adhesive actuators which can pick-and-place flat and flexible materials and also delicate objects such as a light bulb. Despite being an intelligent and shape-adaptive handling system, the gripper design involves complexities due to mechanical architecture and integrated electronic circuitry. Moreover, the gripper failed to estimate the shape and size of the target object and thus needed continuous human cognizance for successful operation. To enhance the grasp on the target object Zhang et al.²² designed gripper jaws based on a set of trapezoidal jaw modules that maximize contact between the jaws and the object at its desired final orientation as it is grasped and avoided jamming. These jaws are constrained to grasp and rotate the object to its desired orientation and achieved a conformal grasp on the object. However, such jaws may easily damage delicate objects in the process as the optimum gripping force which was dependent on the contact locations, were not continuously measured. To ensure safe gripping of delicate objects and to promote user friendliness in lightweight operations, automatized robots were developed to assists humans in various tasks. Such robots utilize human involvement by remotely controlling pick up and drop operations and object manipulations. However, these automated robots are often at a risk of malfunctioning especially when the robots are not programmed to deal to small variations in the target operation which may only seem to be a trivial task for humans. Designing and programming a robot with such human cognitive adaptability can be expensive, complex, and involve complexities in mechanical design and electronic circuitry^{4, 21, 23}. Collaborative robots termed as ‘cobots’ operates alongside human involvement and uses human cognizance to complete a given task effectively^{13, 24}. However, such robot is not suitable for various domestic, household and office task as it requires continuous and appropriate involvement of human cognizance and fails in the purpose for automated pickup and delivery tasks which in turn limits its acceptability. Since robotic applications involves simplification of task for improvement in human lives, highly efficient robots with cognitive decision-making capabilities, will transform and diversify their utilization in, domestic, household, workplaces and other assistive tasks. Thus, to provide a user-cognizance-free automated approach for smart robots we propose the development of ink-jet printed graphene based capacitive multi-sensor array (CAPSENSAR) and utilized for the first time for cognitive decision-making tasks in cognitive robotic gripper (COGBOT), ensuring slippage-free and damage-resistant gripping of the target object.

The drawback of the proposed COGBOT and future work are included in the *Sec. 9-page 25-paragraph 1* and appear in the manuscript as:

However, in spite of various advantages of the COGBOT, it suffers from numerous shortcoming that must be addressed for effective usage. Although the proximity sensor array of the CAPSENSAR offers a high z-axis resolution of 0.091 mm for $z < 30$ mm, it suffers from low x-y flatness resolution of 3 mm and further deteriorates with z. Thus the CAPSENSAR may generate a distorted and blurred capacitive impression of the object when kept at large distances $z > 60$ mm from the device. The x-y flatness resolution of the CAPSENSAR was limited by the elementary sensor unit dimension $a = 3$ mm and the interelectrode spacing $\zeta = 3$ mm and thus features on the object with dimensions < 3 mm may not be clearly distinguished during shape detection by the device unless the features are sharp enough to generate high surface charge density. Although the proposed CAPSENSAR was bendable, the estimation of object shape using a curved device may not yield the right shape of the target object unless the device was recalibrated incorporating the curvature of the device during measurement. During determination of the fittest pair of opposite face of the object prior to gripping, the palm must be parallelly aligned and mutually facing each other to facilitate accurate determination of the P_{grip} to ensure safe gripping. Any deviation in alignment may lead to flawed P_{grip} resulting in slippage. To utilize the motion detection ability of the CAPSENSAR, the future work on the COGBOT will includes working with moving objects. For this purpose, the response time of the detection and the time required for the execution of the iterative cycle in the program must be significantly improved.

2. What is the response time of the authors' prototype? What is the influence of the array size (i,j)? 6 mm and 21 mm are quite small sizes in domestic applications. Could the authors justify why such dimensions were used? is this because of lab restriction or some other reasons?

The authors are pleased to provide a justification to the reviewers' queries. The response time of the CAPSENSAR in COGBOT during proximity and pressure sensing were obtained from Fig. 4 (a) and Fig. 4 (c) (in **page 13** of the revised manuscript) to be 0.3 s and 0.4 s respectively.

The response time of proximity sensor unit is incorporated in the new manuscript and appear in **Sec. 7.1.1 -page 14-paragraph 3** as:

The response time of the proximity sensor unit was graphically determined from Fig. 4 (a) to be 0.3 s. The low response time may be attributed to electrostatic working mechanism of the sensor and enables high switching rate for rapid response sensors for fast generation of capacitive impressions of the exposed face of the object.

The response time of pressure sensor unit is incorporated in the new manuscript and appear in **Sec. 7.1.2 - page 15-paragraph 3** as:

The response time and instability error δ_{stab} of pressure sensor unit was graphically obtained from Fig. 4(c) to be 0.4 s and 5.3% respectively.

The influence of array size (i, j) on the device performance are discussed below:

For the output capacitance of the (i, j) sensor unit of CAPSENSAR to be in the measurable range ~hundreds of fF, the architecture of the device and dimensions of the printed graphene electrodes are optimized to serve the purpose. With graphene-based electrodes of dimensions and the interelectrode distance $a, \zeta < 3$ mm, the output capacitance of the sensor unit is reduced which affects the dynamic range of the proximity sensor. However, when the $a, \zeta > 3$ mm, the sensor unit's baseline reading as well as the dynamic range significantly improves, but compromises the x-y lateral resolution of the CAPSENSAR. Thus, the performance of the CAPSENSAR depends on the a, ζ . However, for large array sized devices, since the performances of the proximity and pressure sensor units were independent in their operation, their role were consistent across the array size ($i \times j$) of the device. *This influence of the of array size (i, j) on the device performance is included in the revised in the Sec. 9 - page 25- paragraph 1 and appear in the manuscript as:*

However, in spite of various advantages of the COGBOT, it suffers from numerous shortcoming that must be addressed for effective usage. Although the proximity sensor array of the CAPSENSAR offers a high z-axis resolution of 0.091 mm for $z < 30$ mm, it suffers from low x-y flatness resolution of 3 mm and further deteriorates with z. Thus the CAPSENSAR may generate a distorted and blurred capacitive impression of the object when kept at large distances $z > 60$ mm from the device. The x-y flatness resolution of the CAPSENSAR was limited by the elementary sensor unit dimension $a=3$ mm and the interelectrode spacing $\zeta=3$ mm and thus features on the object with dimensions < 3 mm may not be clearly distinguished during shape detection by the device unless the features are sharp enough to generate high surface charge density

Justification of the use of the proposed dimensions of the CAPSENSAR:

The CAPSENSAR was designed with restricted electrode area of 5×4 array for prototyping to investigate the technical performances of the device and its trade off with its economic feasibility as it uses printed graphene electrodes in devices and aimed for lightweight household applications. Although the prototype was fabricated with limited active sensing area with five TE and four BE it may be extended to $(2^n \times 2^n)$ sensor array with higher sensing area. Since this work aims to diversify robotic application in household application, the CAPSENSAR was designed for grasping low sized objects so that COGBOT could be more acceptable for a larger role in different sectors including corporate, academia, healthcare, surgery etc in future.

3. The authors showed significant differences in responses to distinct shapes of objects. What are the resolutions of flatness and surface morphology of the prototype?

The authors are thankful to the reviewers for pointing out this. The resolution in flatness (x - y plane) is equal to the dimension of elementary sensor unit and the interelectrode distance $a, \zeta = 3$ mm when the object was in contact with the device $z=0$. Since the x - y error in measurement of the CAPSENSAR was 0.2 mm for every $z=1$ mm, the flatness resolution \mathfrak{R} at a distance z from the device is $3+(0.2z)$. Thus, the flatness resolution of the device deteriorates as the object was placed further away from the device. This is due to the diverging fringing electric field which introduced error in measurement. The z -resolution \mathfrak{R}_z of the device within $z < 30$ mm was obtained from the Figure 4 (a) and (b) (in **page 13** of the revised manuscript) to be 0.091 mm. However, at higher distances $z > 120$ mm the z -resolution was found to be in the order of tens of mm.

The z -resolution \mathfrak{R}_z is included in the **Sec. 7.1.1- page 15-paragraph 2** of the revised manuscript as:

The z -resolution \mathfrak{R}_z of the device within $z < 30$ mm was obtained from the Fig. 4 (a) and (b) to be 0.091 mm. However, at higher distances $z > 120$ mm the z -resolution was found to be in the order of tens of mm. The high stability and excellent z -resolution in the range 0-30 mm make the device suitable for use in surface landscape detection within $z=30$ mm.

The x - y flatness resolution is included in **Sec. 7.2-page 18-paragraph 1** of the revised manuscript as:

The x - y flatness resolution depends on the dimension of electrode size and the interelectrode distance $a, \zeta = 3$ mm. Since the accuracy in the x - y measurement was 80%, the flatness resolution \mathfrak{R} at a distance z from the device is $3+(0.2z)$. The flatness resolution of the device deteriorates as the object was placed further away from the device and was attributed to the diverging fringing electric field of the device.

4. *How do the object materials (e.g., electric/dielectric properties) impact the capacitance and, therefore, the performance of the palm? They should be considered in the theoretical analysis, simulation, and experiments.*

The authors thank the reviewers for their suggestion. The effect of material (metal /dielectric) of the object on the output capacitance of the CAPSENSAR are newly included in **Sec. 3.1-page 5-paragraph 3-5, page 6-paragraph 1-5 and page 7-paragraph 1-2** of the revised manuscript and **Sec. 1.2-page 4-paragraph 3, page 5-paragraph 1-3** of the Supplementary sheet. Since the performance of the CAPSENSAR was investigated using custom made metallic stainless-steel material, emphasis is given to metallic object in the theoretical study provided in the manuscript to make it more focused and compact yet detailed. The results of theoretical study (**Sec. 3.1-page 5-paragraph 3-5, page 6-paragraph 1-5 and page 7- paragraph 1-2** of the revised manuscript) were validated by Simulation investigations (**Sec. 4.1-page 9-paragraph 2, Sec. 4.2-page 10-paragraph 2 and Sec. 4.3-page 10-paragraph 3** of the revised manuscript) and confirmed by experimental data (**Sec. 7.1.1-page 15-paragraph 1,2, Sec. 7.1.2-page 16-paragraph 1,2 and Sec. 7.2-page 16-paragraph 3** of the revised manuscript) for metallic object. However separate case for dielectric material as the target object is discussed in **Sec. 1.2-page 4-paragraph 3, page 5-paragraph 1-3** of the Supplementary sheet. The striking differences between the output capacitances of the device exposed to metallic and dielectric object are also discussed through theoretical studies. While in operation of the COGBOT in grasping a given object (metallic/ dielectric), the device should be re-calibrated for objects of metallic or dielectric behavior. To achieve high accuracy in x - y dimensions and z -depth in the z -contour plots, the resistivity (for metallic objects) and dielectric permittivity (for dielectric object) must be taken into account.

The theoretical study involving the effect of approaching metallic object at distance z on output capacitance of sensor unit is given in **Sec. 3.1-page 5-paragraph 3-5, page 6-paragraph 1-5 and page 7- paragraph 1-2** of the modified manuscript and appear as:

The CAPSENSAR consists of mutually orthogonal arrangement of TE and BE at separate planes to form an arrangement of 5×4 capacitive proximity and pressure sensor array. The proximity sensor of the CAPSENSAR works on the principle of distortion in fringing electric field lines, which emanates from positively biased TE and terminates at negatively biased BE, when an object is introduced in its vicinity (Fig. 1(c)). On the other hand, the pressure sensor works on the change in effective dielectric thickness of CAPSENSAR under applied pressure P . The device is mathematically investigated by analysing the performance of an arbitrary sensor unit in the array. To investigate the performance of an arbitrary (i, j) elementary sensor unit in the CAPSENSAR, mathematical modelling was performed on that sensor unit by considering the influences due to the nearest and the second nearest neighbouring sensor unit. Theoretical studies was carried out to determine the change in output capacitance ΔC experienced by the (i, j) sensor unit in presence of an approaching metallic object along the normal to that sensor unit at proximal distance z from it as discussed in detailed in Supp sheet Sec. 1.1.

If we assume that $C_{out}^{i,j}]^{abs}$ is the output capacitance between the sE of TE and BE of a (i, j) elementary sensor unit without proximal object and $C_{out}^{i,j}]^{pre}$ is the output capacitance as recorded by the same sensor unit in presence of the object at z , then change in output capacitance in terms of the proximal distance z is:
 $\Delta C_{PROX} = C_{out}^{i,j}]^{abs} - C_{out}^{i,j}]^{pre} \dots\dots(1)$

Again, the ΔC_{PROX} can be expressed as: $\Delta C_{PROX} = \frac{Q_{obj}}{-\Delta V_{i,j}} = \frac{(\sigma_{TE} - \sigma_{BE}) \cdot A}{(E_{fr}^{i,j} - E_{fr}^{obj}) \cdot z} \dots (2)$ (See Supp. Sheet Sec. 1.1).

Where, $\Delta V_{i,j}$ denotes the change in potential drop across the TE and BE of the (i, j) sensor unit, E_{fr}^{obj} is the non-uniform fringing electric field between the object and the TE due to the charges Q_{obj} induced at the surface of the object and has an arctan dependence with z (Fig. 2 (b-i)), $E_{fr}^{i,j}$ intrinsic electric field between TE and BE when the object is absent, σ_{TE} and σ_{BE} are surface charge densities of TE and BE electrodes respectively and $(\sigma_{TE} - \sigma_{BE}) \cdot A$ is material dependent quantity since $(\sigma_{TE} - \sigma_{BE}) \cdot A = \sigma_{obj} \cdot S$, where σ_{obj} denotes the surface charge density on the object of surface area S , A effective area of sensor unit of side a . For any z , since $E_{fr}^{i,j} > E_{fr}^{obj}$, the $C_{out}^{i,j}]^{pre} < C_{out}^{i,j}]^{abs}$ according to Eq. 1 and Eq. 2.

The change in output capacitance ΔC_{PROX} of the (i, j) proximity sensor unit is calculated by considering non-overlapping square geometry of sE of TE and BE in parallel capacitive arrangement as :

$$\Delta C_{PROX} = \frac{(\sigma_{TE} - \sigma_{BE}) \cdot A}{z \cdot \left[E_{fr}^{i,j} - \frac{\sigma_{TE}}{\pi \epsilon} \tan^{-1} \left(\frac{a^2}{2z\sqrt{2a^2 + 4z^2}} \right) \right] \left[1 + 4 \left(\frac{z}{\sqrt{z^2 + \zeta^2}} \right) + 4 \left(\frac{z}{\sqrt{z^2 + 2\zeta^2}} \right) \right]} \dots\dots\dots(3)$$

Where ϵ and ζ are the absolute dielectric permittivity of the medium (air) and interelectrode distance between two sub electrodes of TE respectively.

The ΔC_{PROX} increases sharply when the object approaches $z \rightarrow 0$, while the function decays to zero at large $z \rightarrow \infty$ to yield $C_{out}^{i,j}]^{pre} \rightarrow C_{out}^{i,j}]^{abs}$. Since the charge density on the surface of a metallic object (conductor) with finite conductivity depends on the skin depth of the metal at a fixed frequency, the σ_{obj} is dependent on the metallic property of the object. Thus the ΔC_{PROX} varies when the experiment is performed with object of different metal having same shape and size. Although the σ_{obj} is a shape dependent quantity, the ΔC_{PROX} of the single sensor unit is affected when the dimension of the local curvature $< a=3$ mm. Thus for objects of dimensions $\geq a$, exposed to the sensor unit at a fixed z , may be considered as planar for which ΔC_{PROX} is independent of shape and size of the object and solely depends on z . However for an array of sensor units, each sensor unit records their respective ΔC_{PROX} to construct a capacitive landscape of the segmented face of the object. The dimensions and surface morphology variations of the object can be computed from the calibration curve of the proximity sensor array. Thus, metallic objects (with dimensions 30×24 mm) of various shapes and sizes can be distinguished from capacitive impressions generated by the CAPSENSAR. On the contrary, when a dielectric material of permittivity ϵ_r is introduced at a proximal distance z from the (i, j) sensor unit,

the $C_{out}^{i,j}]^{pre} > C_{out}^{i,j}]^{abs}$ as the effective dielectric thickness of the device decreases.

At $z=0$ (touch) when the metallic object touches the TE of the sensor unit, the capacitance between TE and the object is annulled and the $C_{out}^{i,j}]^{pre}$ reduces to: $C_{out}^{i,j}]^{pre}_{z \rightarrow 0} = C_{out}^{i,j}]^{abs} - \max|\Delta C_{PROX}| \dots\dots\dots(4)$,

where $\max|\Delta C_{PROX}| \neq \infty$ is a constant for a fixed material and obtained due to roughness of object surface in contact with the TE. Under this condition, the pressure sensor unit records the applied P which commences from the pressure sensing baseline as given by $C_{out}^{i,j}]^{pre}_{z \rightarrow 0}$ in (4), where $C_{out}^{i,j}]^{abs} = \frac{\sigma_{TE}A}{E_{fr}^{i,j} \cdot d_{fr}|_{z=0}} \dots\dots(5)$ and d_{fr} is the effective

dielectric thickness of the device in the presence of the object at $z=0$. The $C_{out}^{i,j}]^{pre}_{z \rightarrow 0}$ is a constant quantity for a particular object, obtained under just contact condition $z=0$. The effective dielectric layer constitutes the parallelly arranged Ecoflex and PI of thicknesses d_{PI} and δ respectively. Under applied pressure P , the decrease in thickness of the eco-flex elastomeric dielectric layer reduces the d_{fr} which linearly increases the $C_{out}^{i,j}]^{pre}$ with applied pressure P on the sensor unit as:

$$C_{out}^{i,j}]^{pre} = \frac{\sigma_{TE}A.P}{E_{fr}^{i,j} \cdot K} - \max|\Delta C_{PROX}|, \dots\dots\dots (6)$$

$$\Delta C_{PRES} = C_{out}^{i,j}]^{pre} - C_{out}^{i,j}]^{pre}_{z \rightarrow 0} = \frac{\sigma_{TE}A.P}{E_{fr}^{i,j} \cdot K} - \frac{\sigma_{TE}A}{E_{fr}^{i,j} \cdot d_{fr}|_{z=0}} \dots\dots\dots (7)$$

where $d_{fr} = K.P$ and K proportionality constant. The ΔC_{PRES} increases linearly with pressure P with slope $\frac{\sigma_{TE}A}{E_{fr}^{i,j} \cdot K}$ as the second term in (7) is a constant for a fixed object.

The theoretical study involving the effect of approaching dielectric object at distance z on output capacitance of sensor unit is given in *Sec. 1.2 - page 4-paragraph 3, page 5-paragraph 1-3 of the Supp sheet*. and appear as:

Figure S2: Schematic representation of a dielectric object placed at a proximal distance z from a (i, j) unit of CAPSENSAR

To determine the change in output capacitance of the (i, j) sensor unit of the CAPSENSAR due to an approaching object at proximal distance z as shown in Fig. 2, we consider the object of thickness τ , relative dielectric constant ϵ_r approaching the (i, j) sensor unit along its normal direction of $(-z)$.

The output capacitance measured by the (i, j) sensor unit is given by $C_{out}^{i,j}] = \frac{Q}{V_{i,j}}$,

Where Q is the charge on TE and $V_{i,j}$ is the potential drop across TE and BE. In the absence of the object the output capacitance is given by $C_{out}^{i,j}]^{abs} = \frac{Q}{E_{fr}^{i,j} \cdot d_{fr}(z)}$. When the dielectric object is exposed to the (i, j) sensor unit of the

CAPSENSAR in air of $\epsilon_r=1$, the output capacitance $C_{out}^{i,j}]^{pre}_{die} = \frac{Q}{V_{i,j}}$ as the potential drop $V_{i,j}$ across TE and BE

decreases. The $V_{i,j}$ between TE and BE in presence of the object is calculated using the relation: $V_{i,j} = (d_{fr}(z) - \tau) E_{fr}^{i,j} + \frac{E_{fr}^{i,j}}{\epsilon_r} \tau$, where $E_{fr}^{i,j}$ is the intrinsic fringing field between TE and BE and $d_{fr}(z) = \Theta.z$ is the effective dielectric thickness of the sensor unit for $z > 0$, where Θ is the linear proportionality function of z .

$$\text{Thus } V_{i,j} = (\Theta.z - \tau) E_{fr}^{i,j} + \frac{E_{fr}^{i,j}}{\epsilon_r} \tau$$

$$V_{i,j} = (\Theta.z - \tau) E_{fr}^{i,j} + \frac{E_{fr}^{i,j}}{\epsilon_r} \tau$$

$$V_{i,j} = E_{fr}^{i,j} \left[\Theta.z - \tau + \frac{\tau}{\epsilon_r} \right]$$

$$V_{i,j} = E_{fr}^{i,j} \left[\Theta.z - \tau \left(1 + \frac{1}{\epsilon_r} \right) \right]$$

Output capacitance of the (i,j) sensor unit in presence of the object is given by

$$C_{out}^{i,j}]^{pre}_{die} = \frac{Q}{V_{i,j}} = \frac{\sigma_{TE} \cdot A}{E_{fr}^{i,j} \left[\Theta.z - \tau \left(1 + \frac{1}{\epsilon_r} \right) \right]}$$

Unlike metallic object, when a dielectric material of permittivity ϵ_r is introduced at a proximal distance z from the (i,j)

sensor unit, the $C_{out}^{i,j}]^{pre} = \frac{Q}{E_{fr}^{i,j} \left[\Theta.z - \tau \left(1 + \frac{1}{\epsilon_r} \right) \right]}$ increases relative to $C_{out}^{i,j}]^{abs} \left(= \frac{Q}{E_{fr}^{i,j} \cdot \Theta.z} \right)$ as the $\Theta.z$ decreases

by $\tau \left(1 + \frac{1}{\epsilon_r} \right)$ on introduction of the dielectric object. For a fixed object the $C_{out}^{i,j}]^{pre}$ varies linearly with z .

5. *The theory part is weak and considered only a sphere. It should be more comprehensive and convincing to support the simulation and experiments.*

The authors thank the reviewers for their suggestions. The theory of the CAPSENSAR including the mathematical analysis was re-established in the modified theory section. The calibration curve for an elementary sensor unit of the CAPSENSAR was mathematically derived considering a section of the CAPSENSAR, the results were validated using simulation and established using experimental data. The theory was rewritten and appear in the newly modified manuscript in **Sec. 3.1 and page 5-paragraph 3-5, page 6-paragraph 1-5 and page 7- paragraph 1-2 of the modified manuscript** as:

Theoretical derivation in **Sec. 3.1 - page 5- paragraph 3-5, page 6-paragraph 1-5 and page 7- paragraph 1-2** of the modified manuscript appears as:

The CAPSENSAR consists of mutually orthogonal arrangement of TE and BE at separate planes to form an arrangement of 5x4 capacitive proximity and pressure sensor array. The proximity sensor of the CAPSENSAR works on the principle of distortion in fringing electric field lines, which emanates from positively biased TE and terminates at negatively biased BE, when an object is introduced in its vicinity (Fig. 1(c)). On the other hand, the pressure sensor works on the change in effective dielectric thickness of CAPSENSAR under applied pressure P . The device is mathematically investigated by analysing the performance of an arbitrary sensor unit in the array. To investigate the performance of an arbitrary (i,j) elementary sensor unit in the CAPSENSAR, mathematical modelling was performed on that sensor unit by considering the influences due to the nearest and the second nearest neighbouring sensor unit. Theoretical studies was

carried out to determine the change in output capacitance ΔC experienced by the (i, j) sensor unit in presence of an approaching metallic object along the normal to that sensor unit at proximal distance z from it as discussed in detailed in Supp sheet Sec. 1.1.

If we assume that $C_{out}^{i,j}]^{abs}$ is the output capacitance between the sE of TE and BE of a (i, j) elementary sensor unit without proximal object and $C_{out}^{i,j}]^{pre}$ is the output capacitance as recorded by the same sensor unit in presence of the object at z , then change in output capacitance in terms of the proximal distance z is:
 $\Delta C_{PROX} = C_{out}^{i,j}]^{abs} - C_{out}^{i,j}]^{pre} \dots\dots(1)$

Again, the ΔC_{PROX} can be expressed as: $\Delta C_{PROX} = \frac{Q_{obj}}{-\Delta V_{i,j}} = \frac{(\sigma_{TE} - \sigma_{BE}) \cdot A}{(E_{fr}^{i,j} - E_{fr}^{obj}) \cdot z} \dots (2)$ (See Supp. Sheet Sec. 1.1).

Where, $\Delta V_{i,j}$ denotes the change in potential drop across the TE and BE of the (i, j) sensor unit, E_{fr}^{obj} is the non-uniform fringing electric field between the object and the TE due to the charges Q_{obj} induced at the surface of the object and has an arctan dependence with z (Fig. 2 (b-i)), $E_{fr}^{i,j}$ intrinsic electric field between TE and BE when the object is absent, σ_{TE} and σ_{BE} are surface charge densities of TE and BE electrodes respectively and $(\sigma_{TE} - \sigma_{BE}) \cdot A$ is material dependent quantity since $(\sigma_{TE} - \sigma_{BE}) \cdot A = \sigma_{obj} \cdot S$, where σ_{obj} denotes the surface charge density on the object of surface area S , A effective area of sensor unit of side a . For any z , since $E_{fr}^{i,j} > E_{fr}^{obj}$, the $C_{out}^{i,j}]^{pre} < C_{out}^{i,j}]^{abs}$ according to Eq. 1 and Eq. 2.

The change in output capacitance ΔC_{PROX} of the (i, j) proximity sensor unit is calculated by considering non-overlapping square geometry of sE of TE and BE in parallel capacitive arrangement as :

$$\Delta C_{PROX} = \frac{(\sigma_{TE} - \sigma_{BE}) \cdot A}{z \cdot \left[E_{fr}^{i,j} - \frac{\sigma_{TE}}{\pi \epsilon} \tan^{-1} \left(\frac{a^2}{2z\sqrt{2a^2 + 4z^2}} \right) \left[1 + 4 \left(\frac{z}{\sqrt{z^2 + \zeta^2}} \right) + 4 \left(\frac{z}{\sqrt{z^2 + 2\zeta^2}} \right) \right] \right]} \dots\dots\dots(3)$$

Where ϵ and ζ are the absolute dielectric permittivity of the medium (air) and interelectrode distance between two sub electrodes of TE respectively.

The ΔC_{PROX} increases sharply when the object approaches $z \rightarrow 0$, while the function decays to zero at large $z \rightarrow \infty$ to yield $C_{out}^{i,j}]^{pre} \rightarrow C_{out}^{i,j}]^{abs}$. Since the charge density on the surface of a metallic object (conductor) with finite conductivity depends on the skin depth of the metal at a fixed frequency, the σ_{obj} is dependent on the metallic property of the object. Thus the ΔC_{PROX} varies when the experiment is performed with object of different metal having same shape and size. Although the σ_{obj} is a shape dependent quantity, the ΔC_{PROX} of the single sensor unit is affected when the dimension of the local curvature $< a=3$ mm. Thus for objects of dimensions $\geq a$, exposed to the sensor unit at a fixed z , may be considered as planar for which ΔC_{PROX} is independent of shape and size of the object and solely depends on z .

However for an array of sensor units, each sensor unit records their respective ΔC_{PROX} to construct a capacitive landscape of the segmented face of the object. The dimensions and surface morphology variations of the object can be computed from the calibration curve of the proximity sensor array. Thus, metallic objects (with dimensions 30×24 mm) of various shapes and sizes can be distinguished from capacitive impressions generated by the CAPSENSAR. On the contrary, when a dielectric material of permittivity ϵ_r is introduced at a proximal distance z from the (i, j) sensor unit,

the $C_{out}^{i,j}]^{pre} > C_{out}^{i,j}]^{abs}$ as the effective dielectric thickness of the device decreases.

At $z=0$ (touch) when the metallic object touches the TE of the sensor unit, the capacitance between TE and the object is annulled and the $C_{out}^{i,j}]^{pre}$ reduces to: $C_{out}^{i,j}]_{z \rightarrow 0}^{pre} = C_{out}^{i,j}]^{abs} - \max |\Delta C_{PROX}| \dots\dots\dots(4)$,

where $\max |\Delta C_{PROX}| \neq \infty$ is a constant for a fixed material and obtained due to roughness of object surface in contact with the TE. Under this condition, the pressure sensor unit records the applied P which commences from the pressure

sensing baseline as given by $C_{out}^{i,j}]_{z \rightarrow 0}^{pre}$ in (4), where $C_{out}^{i,j}]^{abs} = \frac{\sigma_{TE} A}{E_{fr}^{i,j} \cdot d_{fr} |_{z=0}}$ (5) and d_{fr} is the effective dielectric thickness of the device in the presence of the object at $z=0$. The $C_{out}^{i,j}]_{z \rightarrow 0}^{pre}$ is a constant quantity for a particular object, obtained under just contact condition $z=0$. The effective dielectric layer constitutes the parallelly arranged Ecoflex and PI of thicknesses d_{PI} and δ respectively. Under applied pressure P , the decrease in thickness of the eco-flex elastomeric dielectric layer reduces the d_{fr} which linearly increases the $C_{out}^{i,j}]_{z \rightarrow 0}^{pre}$ with applied pressure P on the sensor unit as:

$$C_{out}^{i,j}]_{z \rightarrow 0}^{pre} = \frac{\sigma_{TE} A \cdot P}{E_{fr}^{i,j} \cdot K} - \max |\Delta C_{PROX}|, \dots \dots (6)$$

$$\Delta C_{PRES} = C_{out}^{i,j}]_{z \rightarrow 0}^{pre} - C_{out}^{i,j}]_{z \rightarrow 0}^{pre} = \frac{\sigma_{TE} A \cdot P}{E_{fr}^{i,j} \cdot K} - \frac{\sigma_{TE} A}{E_{fr}^{i,j} \cdot d_{fr} |_{z=0}} \dots \dots (7)$$

where $d_{fr} = K \cdot P$ and K proportionality constant. The ΔC_{PRES} increases linearly with pressure P with slope $\frac{\sigma_{TE} A}{E_{fr}^{i,j} \cdot K}$ as the second term in (7) is a constant for a fixed object.

Detailed derivation of the ΔC_{PROX} for a (i, j) sensor unit is discussed in **Sec. 1.1-page 1** of supplementary sheet.

The mathematical analysis was supported by simulation studies. The text and Figure in **Sec. 4.1- page 9-paragraph 2 and Sec. 4.3 -page 10-paragraph 3 of the revised manuscript** as:

The determination of the z of the target object (with diameter $\sim a$) from the $(i, j)^{th}$ sensor unit of the CAPSENSAR was obtained by measuring the variation in the fringing field capacitance ΔC_{PROX} between TE and BE. The ΔC_{PROX} occurs due to the variation in spatial fringing electric field distribution $\Delta E_{fr}^{i,j} = E_{fr}^{i,j} - E_{fr}^{obj}$ due to an approaching object at z in the neighbourhood of the $(i, j)^{th}$ sensor unit. The effect of the variation in $E_{fr}^{i,j}$ distribution due to the object near the $(i, j)^{th}$ sensor unit was determined by considering E_{fr}^{obj} generated near the device. Theoretical results obtained from (Supp sheet Sec. 1.1) showed that for an approaching object the E_{fr}^{obj} increases as z reduces obeying the arctan relation as shown in Fig. 2 (b-i). Within close proximity of the device the E_{fr}^{obj} sharply increases as $z \rightarrow 0$. As the object approaches the $(i, j)^{th}$ sensor unit, the distortion field E_{fr}^{obj} interfere with the intrinsic $E_{fr}^{i,j}$ distribution of the device to produce a change in the effective $E_{fr}^{i,j}$ at the that sensor unit as shown in Fig. 2 (b-ii, b-iii). The spatial distribution of the distortion field E_{fr}^{obj} at TE of the $(i, j)^{th}$ sensor unit for object at for $z= 30, 15$ and 0 mm are illustrated in Fig. 2(b-iv), (b-v) and (b-vi) respectively. This distortion field $E_{fr}^{obj} \rightarrow 0$ at large distances and becomes high $E_{fr}^{obj} = 200$ V/m as $z \rightarrow 0$. The values of E_{fr}^{obj} for object placed at different z in the range $z=0-180$ mm were obtained from simulation results and was compared with that of the theoretical counterpart as shown in Fig. 2(b-i). The excellent match between the simulation and the theoretical results was indicative of a dependence of E_{fr}^{obj} on $\arctan(z)$ owing to the non-overlapping square SE architecture of TE and BE which offer non-uniform fringing electric field in proximal space. The increased distortion field E_{fr}^{obj} at reduced z , increases the $\Delta E_{fr}^{i,j}$ and hence the $\Delta C_{PROX} = \frac{Q_{obj}}{\Delta E_{fr}^{i,j} \cdot z}$. Thus proximal distance z of the approaching

object can be measured in terms of ΔC_{PROX} . The capacitive architecture of non- overlapping sub electrodes of TE and BE, with optimised distance between adjacent electrodes ζ in an array, yield a detectable proximal distance of $z=120$ mm. An array of such $(i \times j)$ identical sensor units operated separately to create a capacitive impression and hence the z -landscape of

the exposed object face as discussed below. The non-uniform fringing electric field spanned over large z in space facilitates non-contact shape detection of distant object.

Figure 2: (a) Schematic representation of the architecture of the elementary sensor unit of a section of the CAPSENSAR considered for simulation study in (b), (b-i) Variation of simulated E_{fr}^{obj} with proximal distance z (b) COMSOL representation of fringing electric field $E_{fr}^{i,j}$ and E_{fr}^{obj} distribution around TE and BE of the (i, j) sensor unit in the (ii) absence and (b-iii) presence of object at proximal distance z , under the influence of an approaching object from $z=$ (iv) 30 mm, (v) 15 mm and (vi) in contact $z=0$.

Figure 2: (e) COMSOL representation of Eco-flex thickness variation $\Delta\delta$ of (i, j) sensor unit under (i) $P= 0 \text{ kPa}$, (ii) 2 kPa , (iii) 5 kPa and (f) Variation of Ecoflex thickness and its corresponding change in normalised output capacitance with pressure P

The simulation studies were carried out to investigate the effect of applied pressure P on the effective dielectric thickness d_{fr} in the $(i, j)^{th}$ elementary sensor unit of CAPAENSAR. When the object was in contact with the device $z=0$, the effective dielectric thickness $d_{fr}|_{z=0}$ incorporates the PI thickness d_{PI} and Ecoflex thickness δ . Since the d_{PI} is constant, the application of an external P by the object on the $(i, j)^{th}$ sensor unit produced a change $\Delta\delta$ in thickness of the elastomeric Eco-flex dielectric layer. Simulation studies were performed to determine the displacement $\Delta\delta$ in the Ecoflex layer of $(i, j)^{th}$ sensor unit under different applied P in the range 0-5 kPa. The $\Delta\delta$ occurs in the $-z$ direction in terms of the position of the object on the device ($z=0$). Figure 2 (e-i, e-ii, e-iii) shows the COMSOL representation illustrating the $\Delta\delta$ in Ecoflex thickness under $P=0, 2 \text{ kPa}$ and 5 kPa respectively. At increased pressure P , the Ecoflex thickness δ was reduced, which led to the increase in ΔC_{PRES} of the device following Eq. (7). The linear variation of δ and ΔC_{PRES} with P as obtained from simulation results are plotted in Fig. 2(f). The variation in output capacitance due to change in δ under applied pressure was utilised to measure the gripping force to be applied on the target object.

The mathematical analysis and the simulation results are validated by experimental data and appear in *page 13 Fig. 4(b) and Fig. 4(d), Sec. 7.1.1-page 15-paragraph 1,2 and Sec. 7.1.2-page 16-paragraph 1,2* of the revised manuscript as:

The experimental data when fitted into mathematically obtained calibration curves for proximity and pressure sensors, yielded excellent match. The theoretical, simulation and experimental results are plotted in Fig. 4(b) and Fig. 4(d) as shown in

Figure 4(b): Normalised output capacitance $\Delta C_{PROX} / C_{out}^{i,j,abs}$ vs. proximity distance z calibration curve of a (i, j) sensor unit for an object of stainless steel, relative to the results obtained from theoretical and simulation studies

The calibration curve for each (i, j) sensor unit was obtained by fitting the experimental data in the non-linear equation as:

$$\frac{\Delta C_{PROX}}{C_{out}^{i,j,abs}} = \frac{M_1}{z \cdot \left[M_2 - M_3 \times \tan^{-1} \left(\frac{M_4}{z \sqrt{M_5 + 4z^2}} \right) \left[1 + 4 \left(\frac{z}{\sqrt{z^2 + 36}} \right) + 4 \left(\frac{z}{\sqrt{z^2 + 72}} \right) \right] \right]} \dots (8)$$

where $M_1 = 25.7$ V, $M_2 = 343.3$ V/m, $M_3 = 40.1$ V/m, $M_4 = 28.5 \times 10^7$ m² and $M_5 = 4 \times 10^{10}$ m² for stainless steel object as shown in Fig. 4(b). The Eq. (8) obey theoretically calculated Eq. (3) and validated with simulation results in Fig. 4(b).

Figure 4(d): Normalised change in output capacitance $\Delta C_{PRES} / C_{out}^{i,j,pre}$ vs. Pressure P calibration curve of (i, j) sensor unit showing a dead band in the range 0-0.5kPa and its comparison with simulation results

The calibration curve $\Delta C_{PRES} / C_{out}^{i,j} \Big|_{z \rightarrow 0}^{pre} \cdot P$ for the (i, j) sensor unit is obtained by fitting the experimental data in Eq. 9

as:

$$\frac{\Delta C_{PRES}}{C_{out}^{i,j} \Big|_{z \rightarrow 0}^{pre}} = W_1 \times P - W_2 \dots \dots \dots (9)$$

where, $W_1 = 0.0057 \text{ kPa}^{-1}$ and $W_2 = 0.002$ for stainless steel object. The increase in $\Delta C_{PRES} / C_{out}^{i,j} \Big|_{z \rightarrow 0}^{pre}$ with P in dynamic range is attributed to reduction in Ecoflex thickness δ at increased pressure P . Eq.(9) obey the theoretically derived Eq. (7) and validated with simulation result as shown in Fig. 4(d).

6. *The comparison among theory, simulation, and experiments should be added. Why do equation 6 and equation 5 have different forms of relations between capacitance and distance?*

The authors thank the reviewers for their suggestions. The calibration curve of a sensor unit was mathematically obtained for proximity and pressure sensor in terms of change in capacitance with distance z and pressure P from Eq. (3) and (7) (see *Sec. 3.1 -page 6-paragraph 2,3 and page 7-paragraph 1-2 of the modified manuscript*) of revised manuscript as:

A. Proximity sensor unit:

$$\Delta C_{PROX} = \frac{(\sigma_{TE} - \sigma_{BE}) \cdot A}{z \cdot \left[E_{fr}^{i,j} - \frac{\sigma_{TE}}{\pi \epsilon} \tan^{-1} \left(\frac{a^2}{2z \sqrt{2a^2 + 4z^2}} \right) \right] \left[1 + 4 \left(\frac{z}{\sqrt{z^2 + \varsigma^2}} \right) + 4 \left(\frac{z}{\sqrt{z^2 + 2\varsigma^2}} \right) \right]} \dots \dots (3)$$

Where σ_{TE} and σ_{BE} are surface charge densities of TE and BE electrodes respectively and are material dependent quantity

since $(\sigma_{TE} - \sigma_{BE}) \cdot A = \sigma_{obj} \cdot S$, where σ_{obj} denotes the surface charge density on the object of surface area S

A area of elementary sensor unit of side a

ϵ absolute dielectric permittivity of the medium (air)

$E_{fr}^{i,j}$ intrinsic electric field near TE when the object is absent

ς interelectrode distance between two sub electrodes of TE

B. Pressure sensor unit:

$$\Delta C_{PRES} = C_{out}^{i,j} \Big|_{z \rightarrow 0}^{pre} - C_{out}^{i,j} \Big|_{z=0}^{pre} = \frac{\sigma_{TE} A \cdot P}{E_{fr}^{i,j} \cdot K} - \frac{\sigma_{TE} A}{E_{fr}^{i,j} \cdot d_{fr} \Big|_{z=0}} \dots \dots (7),$$

where $d_{fr} = K \cdot P$ and K proportionality constant.

The equations were validated by simulation studies in *Sec. 4.1- page 9-paragraph 2 and Sec. 4.3 - page 10-paragraph 3* of the revised manuscript and later in the experimental studies in *Sec. 7.1.1- page 15- paragraph 1,2 and Sec. 7.1.2-page 16-paragraph 1,2* of the revised manuscript respectively. The experimental data when fitted into mathematically obtained calibration curves for proximity and pressure sensors, yielded excellent match. The theoretical, simulation and experimental results are plotted in Fig. 4(b) and Fig. 4(d) and appear in the *page 13* of revised manuscript as:

Figure 4(b): Normalised output capacitance $\Delta C_{PROX} / C_{out}^{i,j}]^{abs}$ vs. proximity distance z calibration curve of a (i, j) sensor unit for an object of stainless steel, relative to the results obtained from theoretical and simulation studies

The calibration curve for each (i, j) sensor unit was obtained by fitting the experimental data in the non-linear equation as:

$$\frac{\Delta C_{PROX}}{C_{out}^{i,j}]^{abs}} = \frac{M_1}{z \cdot \left[M_2 - M_3 \times \tan^{-1} \left(\frac{M_4}{z \sqrt{M_5 + 4z^2}} \right) \left[1 + 4 \left(\frac{z}{\sqrt{z^2 + 36}} \right) + 4 \left(\frac{z}{\sqrt{z^2 + 72}} \right) \right] \right]} \dots(8)$$

where $M_1=25.7$ V, $M_2=343.3$ V/m, $M_3=40.1$ V/m, $M_4=28.5 \times 10^7$ m² and $M_5=4 \times 10^{10}$ m² for stainless steel object as shown in Fig. 4(b). The Eq. (8) obey theoretically calculated Eq. (3) and validated with simulation results in Fig. 4(b).

Figure 4(d): Normalised change in output capacitance $\Delta C_{PRES} / C_{out}^{i,j}]^{pre}$ vs. Pressure P calibration curve of (i, j) sensor unit showing a dead band in the range 0-0.5kPa and its comparison with simulation results

The calibration curve $\Delta C_{PRES} / C_{out}^{i,j}]^{pre} - P$ for the (i, j) sensor unit is obtained by fitting the experimental data in Eq. 9

as:
$$\frac{\Delta C_{PRES}}{C_{out}^{i,j}]^{pre}} = W_1 \times P - W_2 \dots\dots\dots(9)$$

where, $W_1=0.0057 \text{ kPa}^{-1}$ and $W_2=0.002$ for stainless steel object. The increase in $\Delta C_{PRES} / C_{out}^{i,j} \Big|_{z \rightarrow 0}^{pre}$ with P in dynamic range is attributed to reduction in Ecoflex thickness δ at increased pressure P . Eq.(9) obey the theoretically derived Eq. (7) and validated with simulation result as shown in Fig. 4(d).

Eq. (5) and Eq. (6) of old manuscript are modified as Eq. (3) and Eq. (8) of revised manuscript respectively. Eq. (3) and Eq. (8) of revised manuscript shows same relation between capacitance and proximal distance of the object.

7. *The quality of the presentation could be improved. In Figure 2, a schematic is needed to make clear the structure. It is hard to read now. In Figure 8, the capacitance changes over time. A detailed explanation should be added to the caption, explaining the operation at the changing time points. The manuscript could be more precise and focused, highlighting the innovation of this study.*

The authors thank the reviewers for their suggestions. A schematic figure 2 (a) is incorporated in the manuscript describing the architecture of the representative (i,j) sensor unit exposed to an approaching object, as considered for theoretical, simulation and experimental studies in Sec 3.1 (**page 6-paragraph 2,3 and page 7- paragraph 1-2 of the modified manuscript**), Sec. 4 (**Sec. 4.1- page 9-paragraph 2, Sec. 4.2- page 10-paragraph 2 and Sec. 4.3 -page 10-paragraph 3 of the revised manuscript**) and Sec. 7 (**Sec. 7.1.1-page 14-paragraph 3, Sec. 7.1.2-page 15-paragraph 3 and Sec. 7.2-page 16-paragraph 3 of the revised manuscript**) respectively. The Figures 2 (b) has also been modified to align with the expressions of Eq. 3 and Eq. 7 in the Sec. 3.1-page 6 and page 7 respectively. Figure 2 (e) is modified and illustrated in terms of change in Ecoflex thickness $\Delta\delta$ while Figure 2(f) is newly included in the manuscript to study the effect of applied P on the $\Delta\delta$ and hence the normalised change in output capacitance and appear in the manuscript as :

Figure 2: (a) Schematic representation of the architecture of the elementary sensor unit of a section of the CAPSENSAR considered for simulation study in (b), (b-i) Variation of simulated E_{fr}^{obj} with proximal distance z , (b) COMSOL representation of fringing electric field $E_{fr}^{i,j}$ and E_{fr}^{obj} distribution around TE and BE of the (i, j) sensor unit in the (ii) absence and (b-iii) presence of object at proximal distance z , under the influence of an approaching object from $z=$ (iv) 30 mm, (v) 15 mm and (vi) in contact $z=0$. (c) COMSOL representation of the electric E -field distribution (E_{fr} array impression) on the elementary sensor units of the CAPSENSAR under exposure to solid objects –(i) sphere, (ii) cone, and (iii) disc. (d) COMSOL representation of E -field pattern (E_{fr} array impression) for (i) Face 1 and (ii) Face 2 under different orientations of custom object placed at distance $z=10$ mm from CAPSENSAR depicted in (iii). (e) COMSOL representation of Eco-flex thickness variation $\Delta\delta$ of (i, j) sensor unit under (i) $P=0$ kPa, (ii) 2 kPa, (iii) 5 kPa and (f) Variation of Ecoflex thickness and its corresponding change in normalised output capacitance with pressure P

The Fig. 8(a) and (b) in **page 13** of old manuscript appear as Fig. 4(a) and (b) in **page 13** of revised manuscript) respectively. Again, Fig. 9(a) and (b) in **page 15** of old manuscript appear as Fig. 4(c) and (d) in the modified manuscript respectively. The Fig. 4(a), (b), (c) and (d) are relabelled to provide

clarity to the experimental results and the captions are rewritten to explain the processes during data acquisition in the experiments.

Figure 4: (a) Dynamic measurement of output capacitance $C_{out}^{i,j,pre}$ with time for an approaching object at different distances $z = 0, 5, 10, 30, 60, 90, 120, 180$ mm, recorded by the (i, j) sensor unit of CAPSENSAR (b) Normalised change in output capacitance $\Delta C_{PROX} / C_{out}^{i,j,abs}$ vs. proximity distance z calibration curve of a (i, j) sensor unit for an object of stainless steel, relative to the results obtained from theoretical and simulation studies (c) Dynamic measurement of output capacitance $C_{out}^{i,j,pre}$ with time for different applied pressures $P = 0.1, 0.5, 1, 2, 3, 4$ and 5 kPa, showing alternative pressure and release cycles when the object was approached onto the (i, j) sensor unit from a distance of $z = 180$ mm for each cycle (d) Normalised change in output capacitance $\Delta C_{PRESS} / C_{out}^{i,j,pre}_{z=0}$ vs. Pressure P calibration curve of (i, j) sensor unit showing a dead band in the range $0-0.5$ kPa and its comparison with simulation results

The supporting data including the theory (Sec 3.1-page 3 of revised manuscript), experimental set-up (Sec. 6-page 11 of old manuscript) and anti-slippage (Sec. 8.4-page 23 of revised manuscript) were shifted to Sec. 1-page 1, Sec. 2-page 6 and Sec. 3-page 7 of the supplementary sheet to make the manuscript more focussed towards the principle highlights and innovation of the study.

The major changes in the new manuscript are listed as follows. All changes are highlighted in yellow.

1. The text in Sec. 1, Sec. 3.1, Sec. 4.1, Sec 4.2, Sec 4.3, Sec 5, Sec 7.1.1, Sec 7.1.2, Sec 7.2, Sec 8.4 and Sec. 9 has been modified/rewritten in the new manuscript and are highlighted.

2. Fig. 2 (a), (b-i), (f) and Fig 9 (c-i), (c-ii), (c-iii) are newly added in revised manuscript and Fig. 2, Fig. 4, Fig. 8, Fig. 9 and Fig. 16 of old manuscript are modified in the new manuscript
3. The different figures are merged and removed to reduce the total figure number to 9 as:
 - I. Fig. 2, 3 and 4 of old manuscript are merged into Fig. 2 in new manuscript
 - II. Fig. 8 and Fig. 9 old manuscript are merged into Fig. 4 in new manuscript
 - III. Fig. 11 and Fig. 12 old manuscript are merged into Fig.6 in new manuscript
 - IV. merging of Fig. 14 and Fig. 15 old manuscript into Fig. 8 in new manuscript
 - V. Fig. 6 and Fig. 7 from new manuscript are removed to supplementary sheet
4. Few parts of Sec. 3.1 of revised manuscript and Sec. 6, Fig. 6 and Fig. 7 of the old manuscript has been shifted to the supplementary sheet to make the manuscript more focused on the research outcome.
5. Reference No. 21, 22, 38 and 39 are newly added in the revised manuscript.

REVIEWERS' COMMENTS:

Reviewer #1 (Remarks to the Author):

The authors have fully addressed my comments and I recommend to publish as it is.

Reviewer #2 (Remarks to the Author):

All comments were will reflected in the response letter.I accepted it to be published.

Reviewer #3 (Remarks to the Author):

The questions have been addressed well. Thanks.